# Constrained Reinforcement Learning Using Successor Representations

**Michael Girstl**[*]    *michael.girstl@tu-darmstadt.de*
*Technical University of Darmstadt (TU Darmstadt)*
*Hessian Center for Artificial Intelligence (hessian.AI)*

**Alexander Mattick**    *alexander.mattick@iis.fraunhofer.de*
*Fraunhofer Institute for Integrated Circuits IIS, Fraunhofer IIS*

**Christopher Mutschler**    *christopher.mutschler@utn.de*
*University of Technology Nuremberg (UTN)*
*Fraunhofer Institute for Integrated Circuits IIS, Fraunhofer IIS*

**Reviewed on OpenReview:** *https://openreview.net/forum?id=6zUq7knzwA*

## Abstract

Real-world Reinforcement Learning depends on the ability to formulate safety constraints into a policy. A common way to model such constraints is to introduce an additional cost signal in the Markov Decision Process, which notifies the agent of unwanted behavior independently of the reward signal. Unfortunately, current methods are hard to adapt to changes in the cost function introduced by, e.g., domain shift or obstacles moving over time. The lack of adaptability means that policies are too unflexible to deal with complex real-world conditions. We propose the Safe Deep Successor Representation (SafeDSR), a novel method that allows quick retraining of policies towards new cost structures. SafeDSR extends the Deep Successor Representation (Kulkarni et al., 2016) to Constrained Reinforcement Learning by introducing a single learnable weight matrix to decouple the learned value function across dynamics, rewards, and costs. This matrix can be updated in a supervised manner instead of having to adapt the whole network if the cost structure of the environment changes. We demonstrate this ability in a freely configurable two-dimensional navigation environment and show that our method is competitive on a simple navigation task while being considerably more flexible.[1]

## 1 Introduction

Reinforcement Learning (RL) is a powerful method for complex world control and planning problems. Unfortunately, RL's real-world utility remains limited due to the inability to encode functional and safety constraints into the policy. As safety cannot be naively enforced on a policy level in standard RL formulations, focus has shifted towards an extension of the normal Markov Decision Process (MDP) towards the Constrained Markov Decision Process (CMDP) (Altman, 1999).

To achieve this, the CMDP augments the MDP five-tuple $(\mathcal{S}, \mathcal{A}, P, R, \gamma)$ (Sutton & Barto, 2018) with (potentially multiple) cost functions $C_j(s, a)$ and a cost budget $b_j$ for $j \in \{1, \ldots, J\}$. The optimization problem

---

[*]Work was done during a collaboration between Fraunhofer IIS and FAU Erlangen-Nürnberg.
[1]The source code is available at `https://github.com/mgirstl/safedsr`.

solving the CMDP can then be formulated as

$$\max_{\pi} \mathbb{E}_{p_{\pi}(\tau)} \left[ \sum_{t=0}^{T} \gamma^t R(s_t, a_t) \right] \quad \text{s.t.} \quad \mathbb{E}_{p_{\pi}(\tau)} \left[ \sum_{t=0}^{T} \gamma^t C_j(s_t, a_t) \right] \leq b_j, \text{ for } j \in \{1, \ldots, J\}, \tag{1}$$

where

$$p_{\pi}(\tau) = \mu(s_0) \prod_{t=0}^{T} P(s_{t+1}|s_t, a_t) \pi(a_t|s_t) \tag{2}$$

is the probability of a trajectory $\tau$ under policy $\pi$, and $\mu$ the initial state distribution.

One noticeable limitation of Equation (1) is that the costs $C_j$ are considered fixed. This makes modeling dynamic or changing environments challenging since adapting to a new constraint (e.g., after a robot is moved to a different position) involves retraining the policy from scratch, even if the objective reward $R(s_t, a_t)$ is fixed. The reasons for allowing quick updates to constraints goes beyond the robotics scenario into, for instance, power-grid management (Chatzivasileiadis, 2018) (plants are installed or taken offline) or workforce scheduling (Birgin et al., 2014) (people get hired or skills change).

In this paper, our aim is to solve this problem using the concept of successor representations (Dayan, 1993) to explicitly model the reachability $M(s, a, s')$ of a state $s'$ starting from any state $s$ after action $a$ was executed. This allows for the full decomposition of dynamics and reward/constraints. Changing the constraints (or rewards for that matter) only means re-weighting the reachability with a new cost or reward function, which is considerably easier to train. Our main contributions can be summarized as follows:

- We propose the Safe Deep Successor Representation (SafeDSR), which leverages the decoupling of dynamics and rewards of the Deep Successor Representation (DSR) (Kulkarni et al., 2016) to model the costs of CMDPs independently of their reward structure.

- We provide initial validation of SafeDSR, demonstrating that the decoupling of the value function enables SafeDSR to adapt to changes in the cost distribution within a few environment interactions (Section 5.4), while maintaining performance comparable to established baseline algorithms on our custom two-dimensional constrained grid environment (Section 5.3).

- For this purpose, we develop a freely configurable constrained grid environment with a continuous state space, allowing for changes in the cost distribution during training.

- We introduce the Safe Deep $Q$-Network (SafeDQN) as a baseline and for comparison to SafeDSR, as both are value-based methods that incorporate constraints using Lagrangian relaxation.

- We motivate the use of the Safe Goal Count and the Average Safe Goal Rate as metrics for comparing algorithms in Constrained Reinforcement Learning (CRL), each offering a single, interpretable evaluation metric that avoids the need to balance multiple objectives.

We structure the remainder of the article as follows. After discussing related work (Section 2), we provide background to Lagrangian relaxations of CMDPs, $Q$-learning and the successor representation (Section 3). Next, we will introduce our two methods: First, we showcase a SafeDQN model without using the successor representation. Second, we introduce our SafeDSR to compare against the naive Lagrangian relaxation provided by SafeDQN (Section 4). Finally, we compare our methods against existing CMDP algorithms on a continuous long-context credit assignment grid world challenge (Section 5).

## 2 Related Work

CRL has historically relied on fixed cost and budget formulations. One of the most seminal works on modern Deep Constrained RL is Constrained Policy Optimization (CPO) (Achiam et al., 2017). This extends the popular TRPO algorithm (Schulman et al., 2015) towards handling external constraints. Stooke et al. (2020) consider combining existing methods like PPO or TRPO with a Lagrangian parameter tuned using

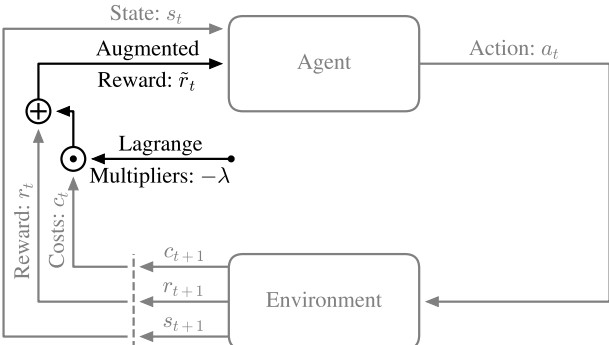

Figure 1: **Agent-environment interface of the CMDP when using Lagrangian relaxation.** By combining the reward $r_t$ and costs $c_t$ using Lagrange multipliers $\lambda$ into the cost-augmented reward $\tilde{r}_t$ (black), the CMDP (gray) appears as a MDP from the agent's perspective. Note that $\lambda$ may change over time.

PID controllers. We follow (Ji et al., 2024) and call these CPPO-PID and TRPO-PID. Interior-point Policy Optimization (IPO) (Liu et al., 2019) considers using interior-point methods to guarantee safe policies remain safe. Reward Constrained Policy Optimization (RCPO) (Tessler et al., 2019) uses a multi-timescale approach as an alternative objective guiding the policy towards the feasible set. Penalized Proximal Policy Optimization (P3O) (Zhang et al., 2022) eliminates constraints by penalizing and clipping the objective to obtain an unconstrained problem. However, while all these methods allow to introduce constraints, they learn to estimate the future expected cost and therefore require full trajectory rollouts to adapt to changes in the cost structure. In contrast, our method separates the dynamics from the cost estimation, allowing updates only to the single-step cost estimator.

On the side of flexibility, the Budgeted Markov Decision Process (BMDP) (Carrara et al., 2019) allows one to change the constraint budget $b$ dynamically without retraining, while keeping the constraint function itself fixed. In contrast, our method is designed to quickly adapt to changes in the cost function $C(s, a)$. This does require some additional training steps (since the new cost $C(s, a)$ can have any arbitrary unknown form), but allows for significantly more flexibility.

The idea of decoupling the rewards from the dynamics to increase the adaptability of RL algorithms is not new. For instance, Weber et al. (2023) proposed a model-based approach that enables adaptation of an objective-conditioned policy at runtime without learning an explicit $Q$-function. This approach relies on learning a reward-free transition model and training the actor network offline by providing an additional differentiable objective-conditioned reward estimator. This has the advantage of additional flexibility, but comes at the cost of explicitly having to model the environment's dynamics. In contrast, our approach is model-free and models the $Q$-function explicitly. This requires the reward predictor to be linear in the learned feature representation, but allows for online training while also being capable of adapting offline within limitations (Note that this is theoretically always possible, see e.g., Hofmann et al. (2008)).

We base our SafeDQN method on the original Deep $Q$-Network (DQN) proposed by Mnih et al. (2015). Our SafeDSR algorithm is based on Deep Successor RL (Kulkarni et al., 2016) and the subsequent work by Barreto et al. (2017). The focus of these two works was dynamic adaptation towards different environments, without considering constraints. This work motivates the usage of such successor representations in CRL.

## 3 Background

### 3.1 Lagrangian Relaxation

As demonstrated in libraries such as OmniSafe (Ji et al., 2024), a common method for finding constraint-satisfying policies $\pi \in \Gamma$ is to formulate the CMDP as a constrained optimization problem and solving it

using Lagrange multipliers $\lambda \in \mathbb{R}_+^J$:

$$\max_\pi \min_{\lambda \geq 0} \mathbb{E}_{p_\pi(\tau)} \left[ \sum_{t=0}^T \gamma^t R(s_t, a_t) - \lambda \cdot \left( \sum_{t=0}^T \gamma^t C(s_t, a_t) - b \right) \right] \tag{3}$$

This formulation allows solving CMDPs using methods designed for solving unconstrained MDPs by training those on cost-augmented rewards $\widetilde{R}(s_t, a_t) = R(s_t, a_t) - \lambda \cdot C(s_t, a_t)$, as depicted in Figure 1, while in parallel updating the Lagrangian parameters using gradient descent. This approach is guaranteed to converge to the optimal policy in the limit of infinite data due to CMDPs having a zero-duality gap (Paternain et al., 2019).

### 3.2 Q-learning

Sequential decision problems can be solved by iteratively updating the state-action value function

$$Q^\pi(s, a) = \mathbb{E}_{p_\pi(\tau)} \left[ \sum_{t=0}^T \gamma^t R(s_t, a_t) \middle| s_0 = s, a_0 = a \right] \tag{4}$$

using temporal-differences and by improving the policy through acting greedily on the current $Q$-value estimates (Watkins & Dayan, 1992). In many practical scenarios, the $Q$-function needs to be approximated using deep neural networks because of too extensive state- and action-spaces. This algorithm is called DQN (Mnih et al., 2015) and uses a target-network and experience replay to stabilize the training.

### 3.3 Successor Representation

Alternatively, we can learn the successor representation

$$M^\pi(s, a, s') = \mathbb{E}_{p_\pi(\tau)} \left[ \sum_{t=0}^T \gamma^t \mathbb{I}(s_t = s') \middle| s_0 = s, a_0 = a \right], \tag{5}$$

which measures the expected (discounted) state occupancy (Dayan, 1993). Here, $\mathbb{I}(x = y)$ is the indicator function, which is 1 if $x = y$ and 0 otherwise. $M^\pi(s, a, s')$ measures how often the state $s'$ is reached when choosing action $a$ being in state $s$. It allows rewriting the $Q$-function using a reward predictor $r(s)$:

$$Q^\pi(s, a) = \sum_{s' \in \mathcal{S}} M^\pi(s, a, s') r(s'). \tag{6}$$

In practice, the successor representation needs to be approximated using deep neural networks as well. For this purpose, Kulkarni et al. (2016) proposed to learn a reward predictor

$$R(s) \approx \phi(s) \cdot w_r \tag{7}$$

using features $\phi(s) \in \mathbb{R}^d$ and reward weights $w_r \in \mathbb{R}^d$, where $d$ is the feature dimension. This assumption is without loss of generality, since any component of $\phi(s)$ can learn to replicate the reward $r(s)$. Those features $\phi(s)$ allow rewriting the $Q$-function as

$$Q^\pi(s, a) \approx \underbrace{\mathbb{E}_{p_\pi(\tau)} \left[ \sum_{t=0}^T \gamma^t \phi(s_t) \middle| s_0 = s, a_0 = a \right]}_{:=\text{ successor features } M^\pi(s,a)} \cdot w_r. \tag{8}$$

DSR (Kulkarni et al., 2016) learns features $\phi$ and reward weights $w_r$ using a reward and reconstruction loss:

$$L_{\text{features}} = \underbrace{\left( R(s) - \phi(s) \cdot w_r \right)^2}_{\text{reward loss}} + \underbrace{\left| \left| s - g(\phi(s)) \right| \right|_2^2}_{\text{reconstruction loss}}. \tag{9}$$

Here, $g$ is an inverse feature network. The successor features $M$ are updated using an adapted DQN loss

$$L = \Big(R(s) + \gamma Q_{\text{prev}}(s', a') - Q(s, a)\Big)^2 \approx \Big|\Big|\phi(s) + \gamma M_{\text{prev}}(s', a') - M(s, a)\Big|\Big|_2^2, \tag{10}$$

where the transition $(s, a, s')$ is sampled from a replay buffer, the action $a'$ is the best action selected using the main network, and $M_{\text{prev}}(s', a')$ is the target network. The parameters of the reward, feature, and inverse feature network are updated in a supervised manner and independently of the parameters of the successor features $M$. Additionally, a biased sampling approach is used to oversample seldomly occurring rewards.

## 4 Methods

We propose SafeDSR as our main algorithm for solving CMDPs. In addition, we define SafeDQN as a baseline for comparison with SafeDSR, since DSR (Kulkarni et al., 2016) builds upon DQN (Mnih et al., 2015). Both algorithms combine a value-based method with Lagrangian relaxation to incorporate cost constraints.

### 4.1 Simple Baseline: Safe Deep Q-Network (SafeDQN)

The SafeDQN algorithm trains the $Q$-network on cost-augmented rewards $\widetilde{R}(s_t, a_t) = R(s_t, a_t) - \lambda \cdot C(s_t, a_t)$, where the costs are scaled with Lagrangian parameters $\lambda$, as sketched in Figure 1. Those parameters are tuned with a proportional controller which increases them if the corresponding constraint is violated at the end of an episode and decreases them otherwise. A detailed description of this algorithm can be found in Appendix A and the used parameters, which are based on the DQN implementation by Huang et al. (2022), in Appendix B. The issue with this approach is that the $Q$-network learns both the dynamics and the cost-augmented reward, where the latter depends on the Lagrangian parameters. Therefore, a large change in those parameters requires the whole $Q$-function to be adapted, which is a slow process.

There are multiple possible solutions to this problem. For instance, Schmidt et al. (2022) proposed to learn a $Q$-function for the future expected reward and a future expected cost predictor $K$ and only use the Lagrangian parameters during decision time. However, in this case all predictors learn the dynamics of the environment, which is computationally expensive and unnecessary because the $Q$- and $K$-functions share the same dynamics. Further, even if the dynamics stay constant, changing the cost or reward structure requires complete retraining of the $Q$- and $K$-functions.

### 4.2 Safe Deep Successor Representation (SafeDSR)

Hence, we use a different approach, the SafeDSR. The DSR allows decoupling the reward and the dynamics by using successor features $M$. This formulation of $Q$-learning is predestined to work with Lagrangian relaxation because the cost-augmented reward is a linear combination of rewards and costs. For this purpose, we generally follow Kulkarni et al. (2016) in our DSR implementation with a few modifications to learn safe policies.

In contrast to Kulkarni et al. (2016), we consider the reward to be state-action-dependent rather than just state-dependent. Hence, the features $\phi$ depend on the state-action pairs $(s, a)$. We model this by concatenating the states $s$ with the one-hot encoded actions $a$. In addition to the successor feature network $M$, we also use a target network $\phi_{\text{prev}}$ for our features $\phi$ to address the moving target in Equation (10). This leads to the successor feature loss

$$L = \Big|\Big|\phi_{\text{prev}}(s, a) + \gamma M_{\text{prev}}(\phi_{\text{prev}}(s', a')) - M(\phi(s, a))\Big|\Big|_2^2, \tag{11}$$

which is outlined in Appendix C.[2] Further, we normalize the feature vectors $\phi$ to $||\phi(s, a)||_2^2 = 1$. This ensures more stability in practice, due to a consistent input range for the subsequent networks. Additionally, Kulkarni et al. (2016) implement the bias sampling approach for training the features $\phi$ and the reward predictor using a secondary buffer containing non-zero reward samples. They draw 20% of the samples from

---

[2]Kulkarni et al. (2016) already used the features $\phi$ as input for the successor features $M$.

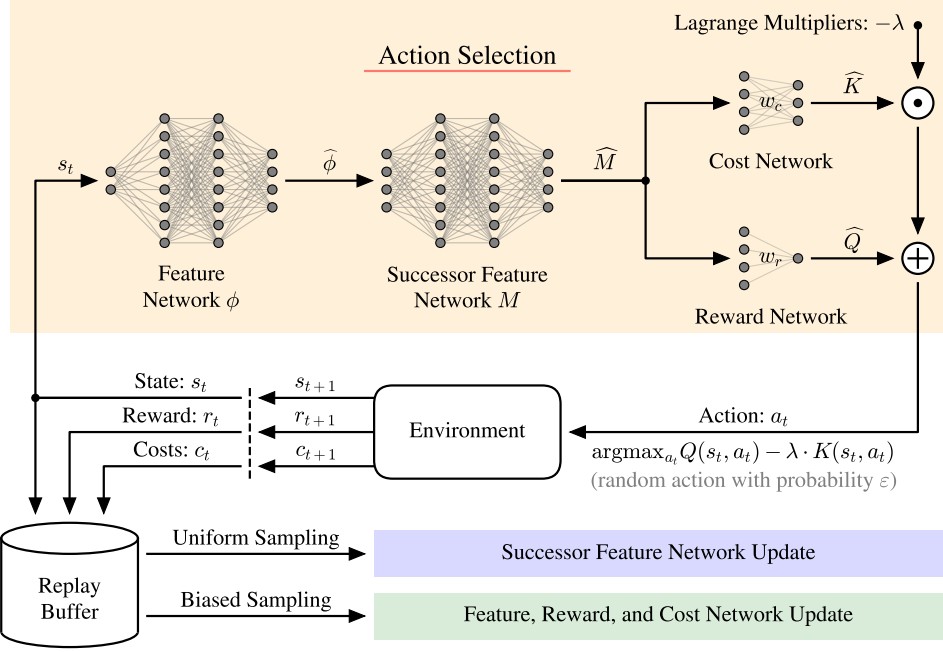

Figure 2: **Overview of the action selection process of the SafeDSR agent.** The agent interacts with the environment, receiving states $s_t$, rewards $r_t$, and costs $c_t$, which are stored in a replay buffer. The different networks are trained by sampling mini batches from the replay buffer. The successor features $M$ are hereby optimized independently of the feature network $\phi$ and the reward and cost networks. The training processes for both are outlined in Appendix C. To select an action $a_t$, the agent calculates the future expected state occupancy $\widehat{M}$ embedded in a high-dimensional space for each possible action $a_t$ in the current state $s_t$. The future expected reward $\widehat{Q} := Q(s_t, a_t)$ and cost $\widehat{K} := K(s_t, a_t)$ are then computed by applying the reward and cost networks to the successor features $\widehat{M}$. This effectively accounts for the reward and costs that occur along the estimated future trajectory. The next action $a_t$ is then selected by choosing the action that maximizes the future expected cost-augmented reward $Q(s_t, a_t) - \lambda \cdot K(s_t, a_t)$, or by choosing a random action with a probability $\varepsilon$. The Lagrange multipliers $\lambda$ are optimized with a proportional controller.

this secondary buffer and 80% from the replay buffer. In contrast, we obtain the 20% by drawing $\frac{20\%}{J+1}$ of samples from all past experiences with probabilities inversely proportional to the reward and to each of the $J$ cost distributions. Finally, we freeze the weights of the feature network $\phi$ after $t_{\text{freeze}}$ environment steps, as proposed by Barreto et al. (2017), which stabilizes the training by fixing the input distribution for the successor network $M$.

To extend the DSR algorithm to work with costs, we add a cost predictor

$$C(s, a) \approx w_c \cdot \phi(s, a), \tag{12}$$

where $w_c \in \mathbb{R}^{J \times d}$ contains the parameters of the cost predictor network. This allows to approximate the future expected costs as

$$K^\pi(s, a) \approx w_c \cdot \underbrace{\mathbb{E}_{p_\pi(\tau)}\left[\sum_{t=0}^{T} \gamma^t \phi(s_t, a_t)\,\middle|\, s_0 = s, a_0 = a\right]}_{:=\text{ successor features } M^\pi(s,a)}. \tag{13}$$

Hence, we need to extend the feature loss Equation (9) to

$$L_{\text{features}} = \underbrace{\beta_r\big(R(s,a) - \phi(s,a) \cdot w_r\big)^2}_{\text{reward loss}} + \underbrace{\beta_c\big|\big|C(s,a) - w_c \cdot \phi(s,a)\big|\big|_2^2}_{\text{cost loss}} + \underbrace{\beta_g\big|\big|(s,a) - g(\phi(s,a))\big|\big|_2^2}_{\text{reconstruction loss}}. \tag{14}$$

Here, $(s, a)$ are the states $s$ concatenated with the one-hot encoded actions $a$, and $\beta_r$, $\beta_c$, and $\beta_g$ are scaling constants (see Appendix B). This equation is also depicted in Appendix C.

Finally, to learn safe policies, we adapt the agent's action selection rule using Lagrangian relaxation:

$$\operatorname*{argmax}_{a} \left\{ Q^\pi(s, a) - \lambda \cdot (K^\pi(s, a) - b) \right\}. \tag{15}$$

An overview of the action selection process is depicted in Figure 2. It differs from the naive Lagrangian relaxation approach in Figure 1, which is used in SafeDQN, by using the Lagrangian parameters only for decision-making, and not for learning future estimates.

We tune the Lagrangian parameters using a proportional controller at the end of each episode, where $\alpha_\lambda$ is the learning rate and $C_{\text{ep}}$ the episode cost. This approach is equivalent to optimizing Equation (3) via gradient descent:

$$\lambda \leftarrow \max(0, \lambda + \alpha_\lambda(C_{\text{ep}} - b)). \tag{16}$$

A detailed description of the SafeDSR algorithm can be found in Appendix A and the used parameters in Appendix B.

## 5 Experiments

In the following, we describe the conducted experiments. In all of them, we use the parameters listed in Appendix B for SafeDQN and SafeDSR. For the other algorithms, we use the implementation and parameters from OmniSafe (Ji et al., 2024), specifically the experimental version developed by Zhou (2023) for discrete action spaces. Further, we always use ten seeds, a discount factor of $\gamma = 0.99$, and a cost budget of 5.

### 5.1 Continuous Grid Environment

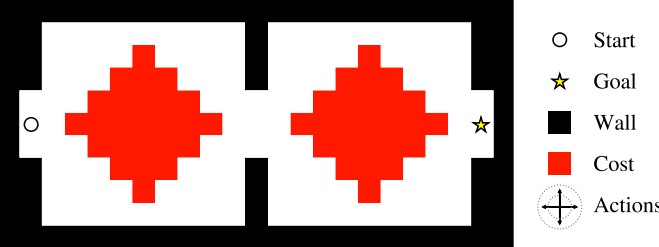

Figure 3: **Constrained grid environment with a continuous state space.** The agent can walk in the cardinal directions with a noisy step length. An episode ends if the agent reaches the goal. Additionally, the agent incurs a cost of 1 for each step taken inside a cost cell.

We consider a goal seeking task in a grid environment, as shown in Figure 3. The dynamics of this environment are based on Barreto et al. (2017) and were extended to a CMDP by adding costs.

The states $s_t$ are the two-dimensional positions of the agent in the maze. They are defined as $s_t = (\alpha x_t, \alpha y_t)$, where $x_t \in [0, x_{\max}] \subset \mathbb{R}$, $y_t \in [0, y_{\max}] \subset \mathbb{R}$ and $\alpha^{-1} = \max(x_{\max}, y_{\max})$. The rescaling with $\alpha$ is done to transfer the state values into a more sensible range for function approximation. The agent can move in the cardinal directions using the discrete actions up, down, left, and right. These actions have a noisy step length drawn from the normal distribution $\mathcal{N}(\mu = 0.75, \ \sigma = 0.075)$, clipped to $\mu \pm 3\sigma$ to simplify collision handling with the walls. If an action sets the agent inside a wall, the action is undone. We use a deterministic reward function where the agent receives a reward of 1 when reaching the goal cell and a time penalty of $-0.01$ for every step taken. This CMDP contains only a single constraint $(J = 1)$, and thus only a single Lagrangian parameter is needed. The agent receives a cost of 1 for each step taken inside a cost cell. The described task ends when the agent reaches the goal. Additionally, we truncate if the agent takes more than 1000

environment steps, since our environment is solvable in fewer than 100 steps, allowing the agent to return to the start if it has gone astray.

It is worth noting that this environment does not fully represent the challenges addressed by modern RL techniques. This environment is similar to common benchmark environments in CRL, for example, the goal task in Safety-Gymnasium (Ji et al., 2023), but with a lower-dimensional state space and discrete actions. The low-dimensionality simplifies plotting of the trajectories and the learned value functions, which facilitates debugging and interpretation. Further, this environment allows for varying the exploration challenge by increasing or decreasing the number of rooms, as shown exemplarily in Appendix D.

## 5.2 Metrics

Typical metrics in CRL include the **future expected reward and cost**, or a variation thereof. The best performing algorithms have a high future expected reward and a low future expected cost at the end of training. However, it is generally unclear and task-dependent whether a slightly larger future expected cost is acceptable for an increase in the future expected reward.

Hence, we consider the **goal and safe goal count**, where the former measures the number of all trajectories during training which reach the goal and the latter only measures trajectories which reach the goal without violating any constraints. Both metrics are monotonically increasing over the training time and more stable than the average episodic reward. We call the derivatives of the goal and safe goal count, the **goal and safe goal rate**. The averages of those rates describe the likelihood of the agent reaching the goal.

Specifically, the safe goal count is defined as

$$\sum_{i=1}^{N_{\text{trajectories}}} \mathbf{1}\Big(\text{goal is reached} \wedge \forall j.C_j(\tau_i) \leq b_j\Big), \tag{17}$$

where $\mathbf{1}(\cdot)$ is the indicator function (1 if the condition is true, 0 otherwise), and $C_j(\tau)$ are the observed costs of trajectory $\tau_i$ for constraint $j$. This acts as a robust proxy for safety, as it discards extreme outliers compared to numerically computing $\mathbb{E}[C_j(\tau)]$. It also allows for meaningful comparison when the environment changes since, while the e.g., absolute rewards might change (since the safe path length changes), the goal count stays comparable.

We focus on these metrics to compare the different algorithms in our experiments. If the metrics are shown as a time series, they are first linearly interpolated to obtain support points every 200 environment steps and then averaged over the different runsusing the interquartile mean for every support point. To obtain a trend line, for instance, the average goal and safe goal rate, we apply a right-aligned window average with a window size of 20 000 environment steps afterwards. If we only discuss a metric at the end of training, we average over the last recorded event using the interquartile mean for each seed. We use the interquartile mean for averaging as it is more robust against outliers (Agarwal et al., 2021).

## 5.3 Baseline Comparison

In this section, we compare SafeDSR with SafeDQN and several baseline algorithms implemented in OmniSafe (Ji et al., 2024). Figure 4 and Figure 5 summarize the main results of this experiment. Additional details such as standard deviations for Figure 4 are included in Appendix F, while Appendix G contains additional details such as standard deviations for Figure 5 as well as time series plots of the average episodic reward and cost. In the one- and two-room environment without costs, many algorithms are capable of reaching the goal more often than SafeDSR. However, when costs are introduced in environments with multiple rooms, only SafeDSR consistently reaches the goal safely with a high average episodic reward and average episodic cost below the cost budget of 5. Because the safe goal count does not show how fast and consistently the algorithms find safe trajectories, we show the average safe goal rate in Figure 6.

In the one-room environment, many algorithms eventually learn safe policies. While the seven best-performing algorithms reliably locate the goal after approximately 250 000 environment steps, their average safe goal rates vary significantly. For example, the PID-Lagrangian methods achieve average safe goal rates

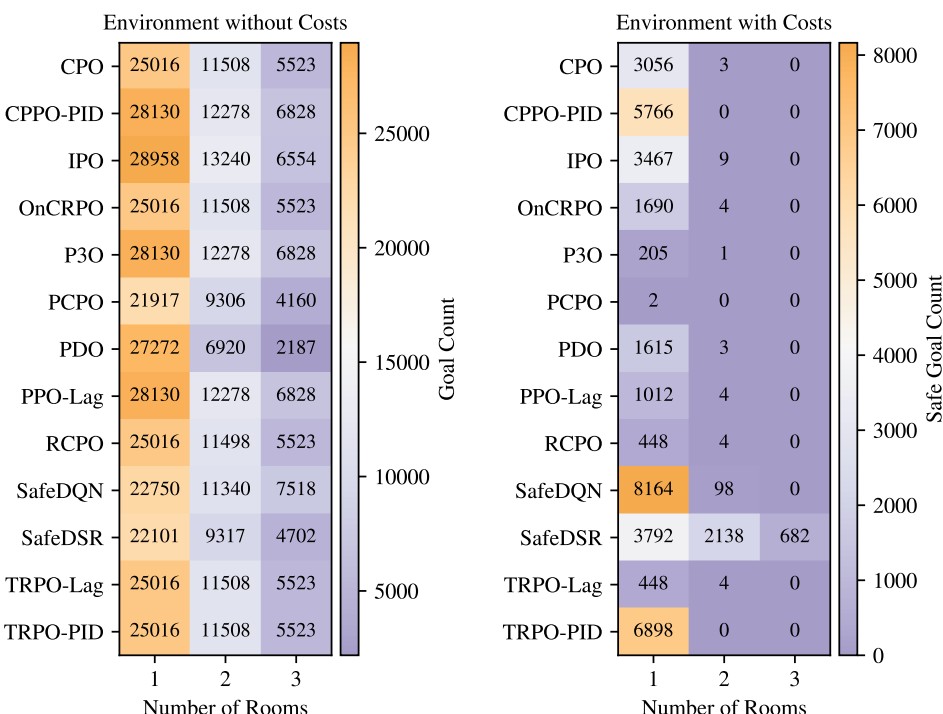

Figure 4: **Goal and safe goal count for the different baseline algorithms, SafeDQN, and SafeDSR** after 500 000 environment steps in environments with and without costs. The counts were averaged over ten seeds and rounded to the nearest integer. On the left, each algorithm was run in the environments, as show in Appendix D without costs. On the right, the same algorithms were used with a cost budget of 5 in the same environments with costs. All algorithms are able to reach the goal in the costless environments. Most are even able to reach a goal count larger than SafeDSR in the environments with one or two rooms. In contrast, only CPPO-PID, SafeDQN, and TRPO-PID outperform SafeDSR in the cost environments when using a single room. In the two- and three-room cost environments SafeDSR outperforms all other algorithms. To maintain readability, additional data such as standard deviations are included in Appendix F.

of over 80% by the end of training, whereas SafeDSR and SafeDQN settle at around 70%. This implies that the naive proportional controller from Equation (16) might be improved by incorporating integral and derivative terms. Other algorithms, such as IPO, show promising performance mid-training but eventually fail to maintain safe trajectories.

The situation differs in the two- and three-room environments, where only SafeDSR reaches an average safe goal rate of over 60% and 30%, respectively. Specifically note that SafeDQN significantly underperforms SafeDSR, despite having an equal exploration strategy and using equal algorithms: The main difference between SafeDQN and SafeDSR is just the parameterization of the $Q$-function as either a "normal" neural network or a Deep Successor Representation. This observation comes with the caveat that SafeDQN can reach a much higher average safe goal rate in the two-room environment given enough training steps. The longer training time for SafeDQN is likely due to the implicit handling of the Lagrangian parameters and costs compared to SafeDSR which explicitly learns the cost distribution. In contrast, we found in additional experiments that even five-times as many environment steps did not improve the performance of the other algorithms significantly. Further, delaying the start of the Lagrangian parameter tuning for the four Lagrangian methods PDO, RCPO, TRPO-Lag and PPO-Lag did only marginally change their performance, compared to SafeDSR where a delay facilitates exploration and improves its performance.

We suspect that many of the tested algorithms are limited by their exploration strategies, which is hinted by the trajectories shown in Figure 7 for the two-room environment. Many algorithms get either stuck in the

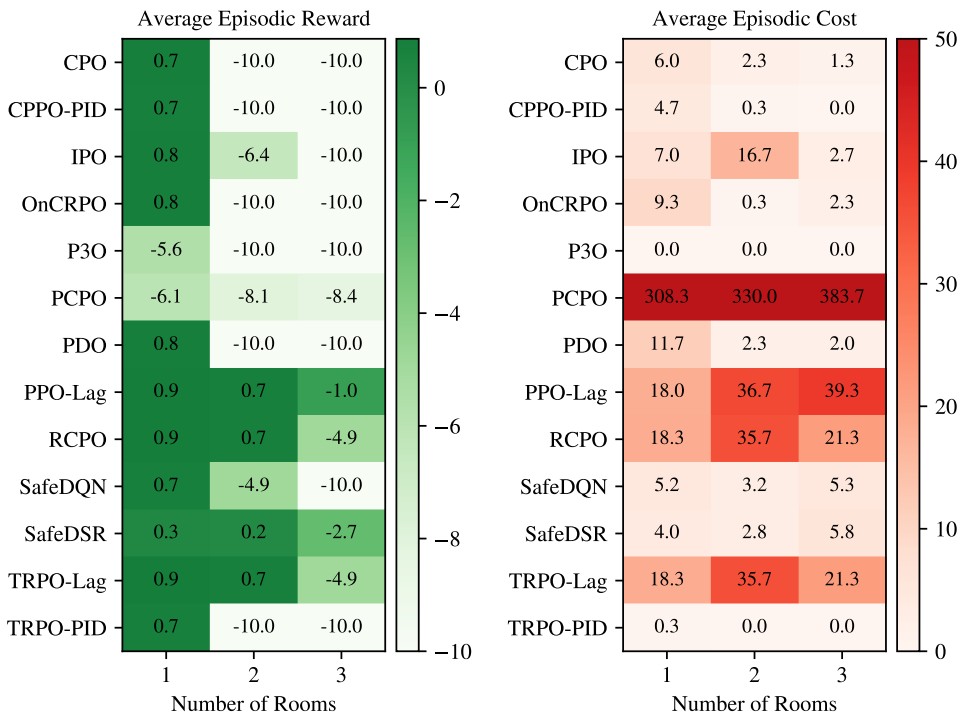

Figure 5: **Average episodic reward and cost for the different baseline algorithms, SafeDQN, and SafeDSR** at the end of training in the environments with costs. In the one-room environment most algorithms are able to reach a higher average episodic reward than SafeDSR while some still maintain an average episodic cost below the cost budget of 5, i.e., CPPO-PID and TRPO-PID. The situation differs in the environments with more rooms, where most algorithms suffer from a trade-off: they either have high average episodic reward and high average episodic cost, or low reward with low cost. Only SafeDSR achieves both low cost and high reward simultaneously. To maintain readability, additional data such as standard deviations are included in Appendix G. Further, the colorbar on the right was clipped to a value of 50 to reduce the influence of outliers.

corners of the environment, because of an overcorrecting behavior towards the cost cells (e.g.,CPPO-PID), or get stuck in local optima (e.g., RCPO). In contrast, both the SafeDSR and SafeDQN agent walk around the cost cells, reaching the goal. This is very likely due to their high exploration rate ($\varepsilon = 0.25$), which increases the likelihood of finding cost-avoiding paths through the second room even if the Lagrangian parameter tuning overcorrected to avoid the cost cells.

We acknowledge that these results are not without limitations, since we use the default parameters for the used OmniSafe algorithms, while the SafeDSR algorithm was tuned for this work and SafeDQN was adapted appropriately. Hence, the OmniSafe implementations might underperform compared to SafeDQN and SafeDSR.

## 5.4 Changing Cost Distribution

Finally, we discuss the model's ability to respond to changing cost distributions. For this, we consider a policy trained on the "diamond" shape shown in Figure 3 and adapt it towards a different shape. We consider three different methods of doing this adaptation: First, one can only change the cost weights $w_c$ without changing the Lagrangian ("post-adaptation" evaluation). This, arguably, is the minimal amount of change necessary to adapt towards an unforeseeable change in cost. In addition, we may also tune the Lagrangian parameter $\lambda$, since a change in cost distribution generally also changes the intensity necessary to repel dangerous states. For this, we adapt just the Lagrangian in an on-policy fashion ("penalty" evaluation). Finally, we consider

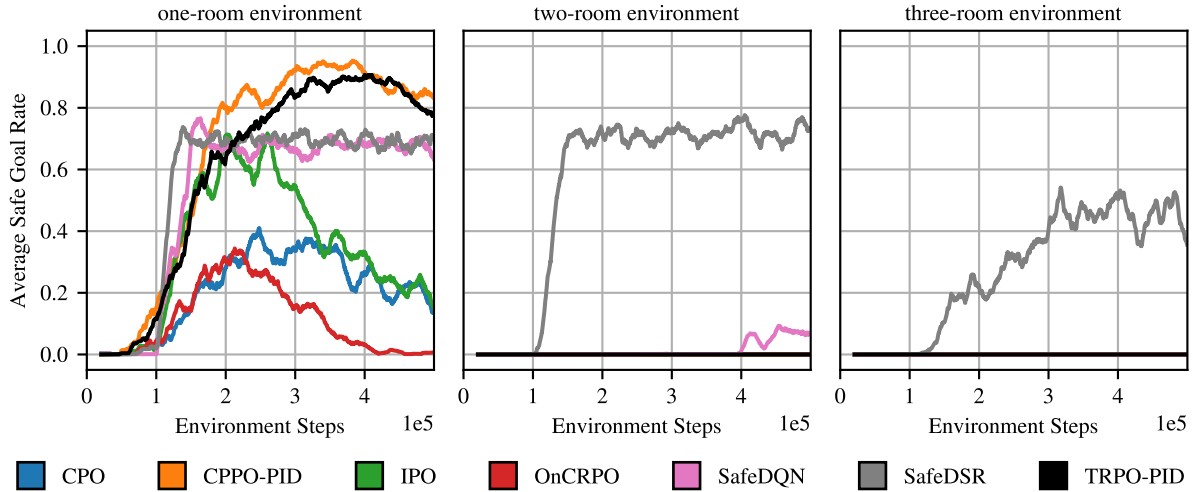

Figure 6: **Average safe goal rate for different baseline algorithms, SafeDQN, and SafeDSR.** We only show the seven best-performing algorithms from Figure 4. The PID-controlled algorithms, CPPO-PID and TRPO-PID have the largest safe goal rates at the end of training in the one-room environment, while SafeDQN and SafeDSR perform slightly worse. In the other environments, only SafeDSR is able to consistently find safe trajectories.

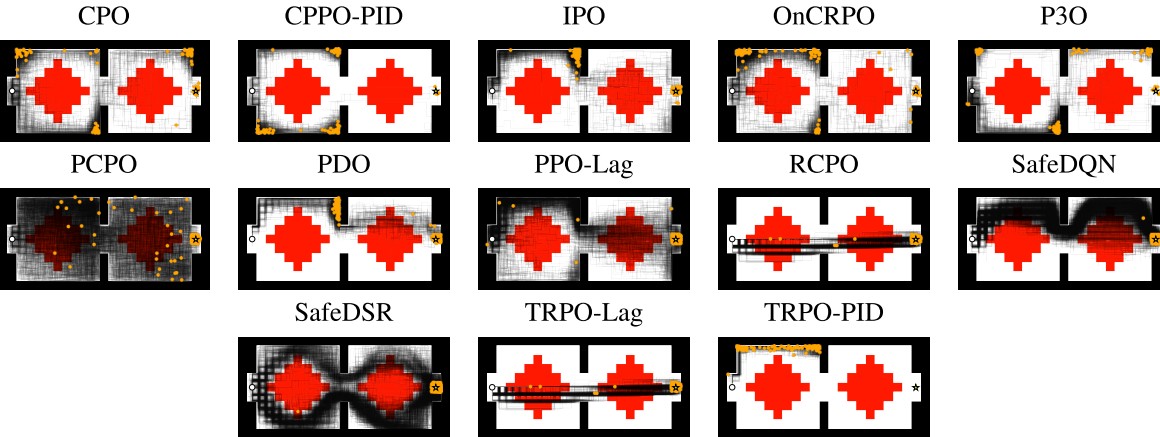

Figure 7: **Trajectories the different baseline algorithms, SafeDQN, and SafeDSR** have observed during the last 100 000 environment steps in the two-room environment. In these plots, we show the trial with the largest safe goal count for each algorithm. We use the goal count as a tie-breaker. Many algorithms are not able to reach the goal safely because they are unable to circumvent the wall separating the rooms. Meanwhile, RCPO and TRPO-Lag got stuck in local optima.

training the full model — including reward, cost, Lagrange parameters, and successor features — for an additional 100 000 gradient steps to demonstrate the theoretically achievable improvement if we didn't take advantage of the factorization into dynamics, reward, and cost models. Note that this is generally necessary if the policy changes too much.

One important consideration when using successor features to adapt towards changing cost and reward functions is that previously uninteresting states may become interesting after the change. Especially in the case of CRL this might lead to states that were entirely unexplored due to being hidden behind the

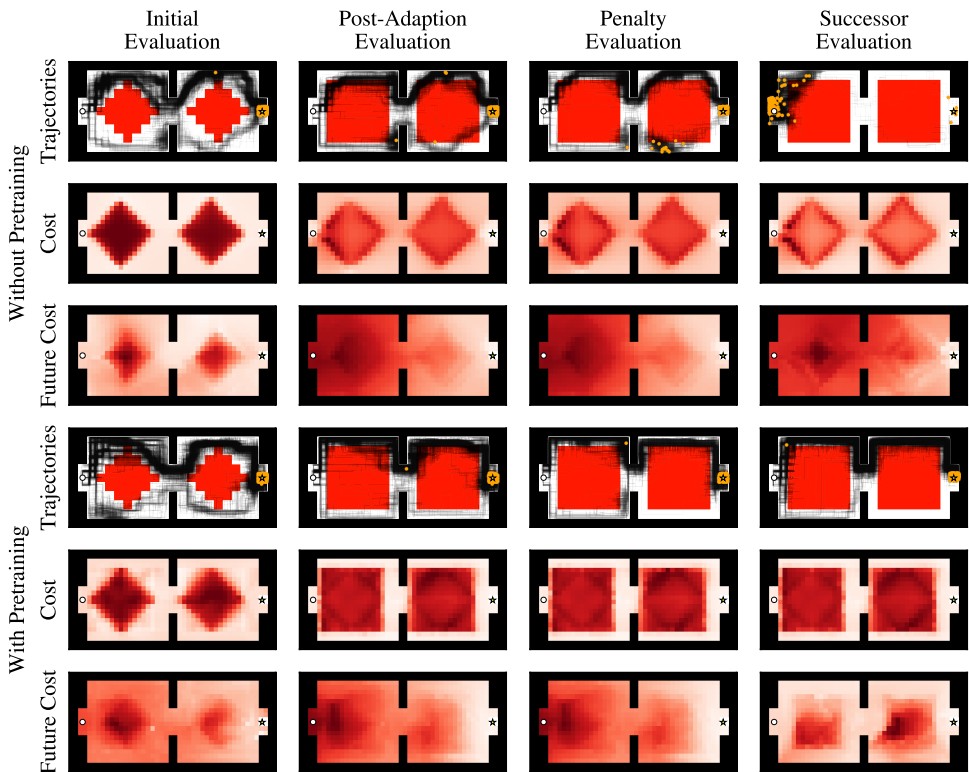

Figure 8: **Translation from a model pretrained on the "diamond" shape (Figure 3) towards an unseen "square" distribution.** We refer to training reward and cost functions as "post-adaptation" evaluation, to training the Lagrangian on-policy as "penalty" evaluation, and finally to training the full network as "successor" evaluation. We also show the performance of our method with or without pretraining the feature embedding model.

constraints. For this reason, we consider a "pretraining" stage where *only* the feature encoder is pretrained using Equation (14) for random rollouts on the continuous grid.[3] In a real-world navigation setting, this would correspond to, e.g., adding random images from an empty warehouse to the dataset to allow for good feature embeddings. We argue that this is a small assumption in practice, as good pretrained models are generally available and can be used as feature embedders (Rahbar et al., 2022).

We showcase the empirical paths walked by our methods similar to Figure 7 since a direct comparison of rewards/costs is rather fraught: Changing the environment cost function can make the underlying problem harder or easier to an arbitrary level, so comparing achieved rewards is not meaningful. We supply the average safe goal rate during the different training phases in Appendix H.

We compare two adaptation settings "Diamond → Square" and "Diamond → Inverse Diamond" where the initial "diamond" shape gets adapted into either a large "square" or gets "inverted" such that all feasible areas become infeasible and vice versa (with small connections towards the start and end, see Figure 9). The first setting is used to test the impact of choosing a harder cost distribution than the initial one (since there is significantly less feasible area), while the second setting is intended to test an "adversarial example" where the configuration is built to force the agent to walk through exactly those areas that were infeasible before.[4]

---

[3]Note that this means we do not have any dynamics information in our dataset. The pretraining information is just a set of random observations!

[4]We provide an additional experiment for a scaled "diamond" in Appendix E.

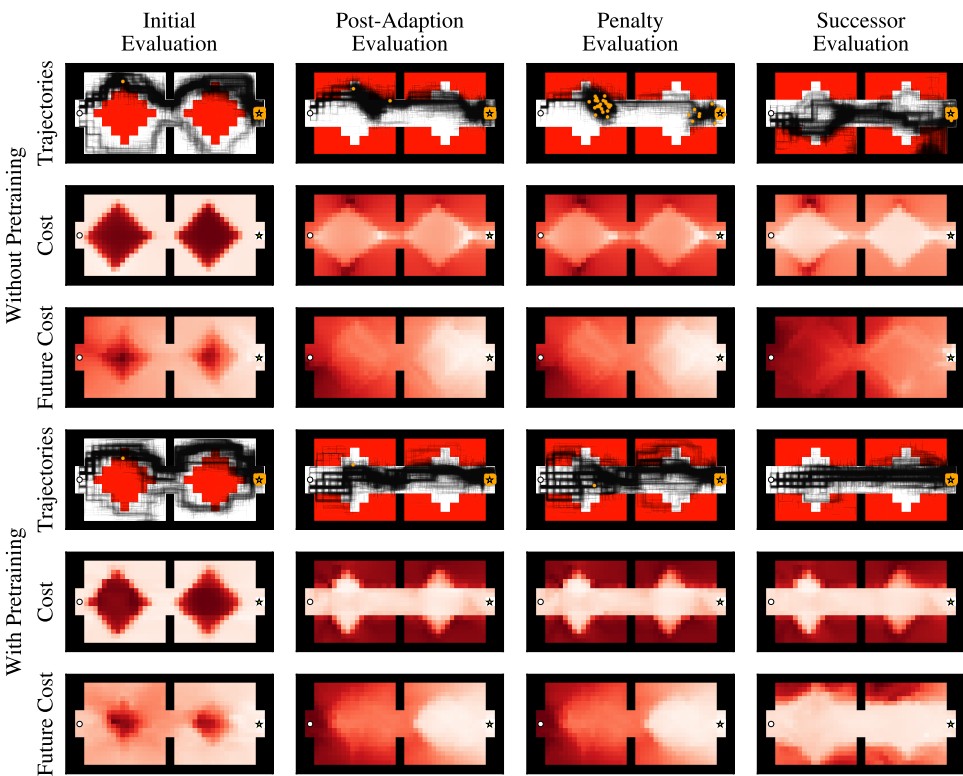

Figure 9: **Translation from a model pretrained on the "diamond" shape (Figure 3) towards an unseen "inverse diamond" distribution.** We refer to training reward and cost functions as "post-adaptation" evaluation, to training the Lagrangian on-policy as "penalty" evaluation, and finally to training the full network as "successor" evaluation. We also show the performance of our method with or without pretraining the feature embedding model.

We show our results in Figure 8, as one can see our method manages to quickly adapt its feasible region towards the new "square" cost structure, even if only the reward and cost function is trained. We also generally see the method with pretraining outperforming the one without, especially once the successor representation itself is trained. This aligns well with prior research on successor representations that highlight the crucial importance of the underlying features.

Looking at the adversarial example case in Figure 9, we find that even in such a worst-case scenario, our method manages to find a feasible policy despite the original policy having the exact opposite movement pattern. We observe that while adapting the cost function itself generally improves the results in all settings, the adaptation of the penalty coefficient and full model training has varying impact: Pretrained models tend to benefit from full model training, while for non-pretrained models performance may actually decrease. We hypothesize that this is due to the fact that there are no good features for the previously constrained areas, which makes the problem partially observable from the point of view of the feature embeddings. The pretrained model has no such dearth of data in the constrained areas, meaning it can learn a good successor model. Nevertheless, both the pretrained and non-pretrained model manage to find solutions to the path finding problem.

# 6 Limitations

One core limitation of our approach is the complexity of scaling Successor Representations to complex environments. Specifically, DSR-based methods tend to struggle in high-dimensional environments where the "features" are not easily linearized (see also Kulkarni et al. (2016); Barreto et al. (2017)): Theoretically,

a linearization always exists via kernel embedding of the transition function, but practically this might be too complex to work with. The typical remedy for this is classical feature engineering. For instance, Barreto et al. (2017) employed grid-based features and Radial Basis Functions (RBFs) to represent their MDP as a linearizable feature set. This way, their state representation is reduced to a relatively sparse feature vector, which simplifies finding DSRs. Kulkarni et al. (2016) directly mention the need for additional objectives (such as unsupervised learning) to scale DSR to realistic problems.

Another limitation of Successor Representations comes with their nature of constructing $Q$-functions: A standard $Q$-learner cannot be used for continuous action spaces since the per-step optimization of $\text{argmax}_a Q(s, a)$ becomes intractable in practice. However, this has already been solved in the form of, e.g., DDPG (Lillicrap et al., 2015), SAC Haarnoja et al. (2018), or TD3 Fujimoto et al. (2018), which all try to amortize the $\text{argmax}_a Q(s, a)$ optimization into a fixed actor. We leave the transfer of our method to those Actor-Critic algorithms for future work.

## 7 Conclusion

We propose a new method for Constrained Reinforcement Learning that allows quick adaptation towards novel cost functions by decomposing the dynamics and the associated costs and rewards. Therefore, our method can respond to unforeseen changes in reward or cost by just changing the reward and cost components, but keeping the dynamics fixed.

We demonstrate our model's ability to act in simple environments by comparing it to existing CRL methods on an interpretable stochastic continuous grid world with freely configurable costs. Our method maintains performance comparable to existing CRL solvers while increasing the flexibility of the trained policy. Finally, we show that our method can learn to adapt to new constraints quickly by adapting a model pretrained on a "diamond" constraint form to different constraint formulations, suggesting the approach shows promise for CRL problems with shifting cost structures.

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

# A Algorithm Pseudocodes

---

Algorithm 1: **Safe Deep $Q$-Network using a proportional controller.**
(Changes from DQN (Mnih et al., 2015) are shown in black.)

---

1: initialize replay buffer $D$
2: initialize action-value function $Q$ with random weights $\theta$
3: initialize target action-value function $Q_{\mathrm{prev}}$ with weights $\theta_{\mathrm{prev}} = \theta$
4: initialize Lagrangian parameters $\lambda = 0$ and episode cost $C_{\mathrm{ep}} = 0$
5: initialize iteration count $t = 0$

6: **for** $N$ episodes **do**
7:     get first state $s$
8:     calculate feature $\phi := \phi(s)$     // $\phi(s)$ can be the identity
9:     **while** $s$ is not terminal **do**
10:         choose $a$ using $\varepsilon$-greedy policy on $Q(\phi, \cdot)$
11:         take action $a$ and observe $r$, $c$, $s'$
12:         update episodic cost $C_{\mathrm{ep}} \leftarrow C_{\mathrm{ep}} + c$
13:         calculate feature $\phi' := \phi(s')$
14:         store transition $(\phi, a, r, c, \phi')$ in $D$
15:         sample random mini batch $(\phi_b, a_b, r_b, c_b, \phi'_b)$ from $D$
16:         **if** $\phi'_b$ terminal **then**
17:             $Q_{\mathrm{target}} := r_b - \lambda \cdot c_b$
18:         **else**
19:             $Q_{\mathrm{target}} := r_b - \lambda \cdot c_b + \gamma \max_{a'} Q_{\mathrm{prev}}(\phi'_b, a')$
20:         **end if**
21:         calculate loss $L = \left(Q_{\mathrm{target}} - Q(\phi_b, a_b)\right)^2$
22:         update $Q$-network using gradient descent $\theta \leftarrow \theta - \alpha \nabla_\theta L$ with learning rate $\alpha$
23:         set $\phi \leftarrow \phi'$ and $t \leftarrow t + 1$
24:         every $k_{\mathrm{sync}}$ steps set $\theta_{\mathrm{prev}} = \theta$
25:         anneal exploration variable $\varepsilon$
26:     **end while**
27:     **if** ($s$ is terminal **or** truncated) **and** ($t > t_{\mathrm{penalty}}$) **then**
28:         update $\lambda \leftarrow \max(0, \lambda + \alpha_\lambda(C_{\mathrm{ep}} - b))$ with learning rate $\alpha_\lambda$
29:         set $C_{\mathrm{ep}} = 0$
30:     **end if**
31: **end for**

---

Algorithm 2: **Safe Deep Successor Representation using a proportional controller.**
(Changes from DSR (Kulkarni et al., 2016) are shown in black.)

1: initialize replay buffer $D$
2: initialize successor features $M$ with random weights $\theta$
3: initialize target successor features $M_{\mathrm{prev}}$ with weights $\theta_{\mathrm{prev}} = \theta$
4: initialize feature network $\phi$ with random weights $\xi$
5: initialize inverse feature network $g$ with random weights $\tilde{\xi}$
6: initialize reward weights $w_r$ randomly
7: initialize cost weights $w_c$ randomly
8: initialize target features $\phi_{\mathrm{prev}}$ with weights $\xi_{\mathrm{prev}} = \xi$
9: initialize Lagrangian parameters $\lambda = 0$ and episode cost $C_{\mathrm{ep}} = 0$
10: initialize iteration count $t = 0$

11: **for** $N$ episodes **do**
12:     get first state $s$
13:     **while** $s$ is not terminal **do**
14:         choose $a$ using $\varepsilon$-greedy policy on $Q(s, \cdot) - \lambda \cdot K(s, \cdot)$
15:         take action $a$ and observe $r$, $c$, $s'$
16:         store transition $(s, a, r, c, s')$ in $D$
17:         update episodic cost $C_{\mathrm{ep}} \leftarrow C_{\mathrm{ep}} + c$
18:         sample random mini batch $(s_b, a_b, r_b, c_b)$ from $D$ using biased sampling
19:         calculate loss $L_{\mathrm{reward}} = \left(r_b - \phi(s_b, a_b) \cdot w_r\right)^2$
20:         calculate loss $L_{\mathrm{cost}} = \left||c_b - w_c \cdot \phi(s, a)\right||_2^2$
21:         calculate loss $L_{\mathrm{reconstruction}} = \left||(s_b, a_b) - g(\phi(s_b, a_b))\right||_2^2$
22:         calculate loss $L_{\mathrm{features}} = \beta_r L_{\mathrm{reward}} + \beta_c L_{\mathrm{cost}} + \beta_g L_{\mathrm{reconstruction}}$ with scaling factors $\beta_r, \beta_c, \beta_g$
23:         **if** $t > t_{\mathrm{freeze}}$ **then**
24:             update $w_r$ and $w_c$ using gradient descent on $L_{\mathrm{features}}$ with learning rate $\alpha$
25:         **else**
26:             update $w_r, w_c, \phi$ and $g$ using gradient descent on $L_{\mathrm{features}}$ with learning rate $\alpha$
27:         **end if**
28:         sample random mini batch $(s_b, a_b, s'_b)$ from $D$ using uniform sampling
29:         calculate next action $a'_b = \mathrm{argmax}_{a'}\left\{Q(s'_b, a') - \lambda \cdot K(s'_b, a')\right\}$
30:         **if** $s'_b$ terminal **then**
31:             $M_{\mathrm{target}} := \phi_{\mathrm{prev}}(s_b, a_b)$
32:         **else**
33:             $M_{\mathrm{target}} := \phi_{\mathrm{prev}}(s_b, a_b) + \gamma M_{\mathrm{prev}}(\phi_{\mathrm{prev}}(s'_b, a'_b))$
34:         **end if**
35:         calculate loss $L = \left||M_{\mathrm{target}} - M(\phi(s_b, a_b))\right||_2^2$
36:         update $M$-network using gradient descent on $L$ with learning rate $\alpha$
37:         set $s \leftarrow s'$ and $t \leftarrow t + 1$
38:         every $k_{\mathrm{sync}}$ steps set $\theta_{\mathrm{prev}} = \theta$ and $\xi_{\mathrm{prev}} = \xi$
39:         anneal exploration variable $\varepsilon$
40:     **end while**
41:     **if** ($s$ is terminal **or** truncated) **and** ($t > t_{\mathrm{penalty}}$) **then**
42:         update $\lambda \leftarrow \max(0, \lambda + \alpha_\lambda(C_{\mathrm{ep}} - b))$ with learning rate $\alpha_\lambda$
43:         set $C_{\mathrm{ep}} = 0$
44:     **end if**
45: **end for**

# B    Parameters

Table 1 contains the SafeDQN and Table 2 the SafeDSR parameters used during the experiments. The SafeDSR algorithm was tuned using the environment depicted in Figure 3 with a fixed Lagrangian parameter $\lambda = 0$, as extensive pre-tuning would hide the actual number of environment steps needed to find a safe solution. For both algorithms, we use the Adam optimizer (Kingma & Ba, 2014) with its default parameters for network training, except for the learning rates.

Table 1: **SafeDQN parameters used during the experiments.** We use the DQN parameters provided by Huang et al. (2022), except the ones used for the proportional controller and the $\varepsilon$-annealing, for which we use the same configurations as for SafeDSR.

| Algorithm Parameters | Value |
|---|---|
| batch size | 128 |
| buffer size | 10 000 |
| discount factor $\gamma$ | 0.99 |
| learning rate $\alpha$ | 0.000 25 |
| total time steps [environment steps] | 500 000 |
| training frequency [environment steps per training step] | 10 |
| training start time [environment steps] | 10 000 |
| target network frequency $k_{\text{sync}}$ [environment steps per synchronization] | 500 |

| $\varepsilon$-Annealing | Value |
|---|---|
| end time [environment steps] | 100 000 |
| end value $\varepsilon_\infty$ | 0.25 |
| start time [environment steps] | 20 000 |
| start value $\varepsilon_0$ | 1 |

| Lagrangian Parameter Tuning | Value |
|---|---|
| cost budget $b$ | 5 |
| learning rate $\alpha_\lambda$ | 0.001 |
| start value $\lambda_0$ | 0 |
| training start time $t_{\text{penalty}}$ [environment steps] | 100 000 |

| $Q$-Network | # Nodes |
|---|---|
| dense layer | 120 |
| ReLU activation function | |
| dense layer | 84 |
| ReLU activation function | |
| dense layer | 4 |

Table 2: **SafeDSR parameters used during the experiments.**

| Algorithm Parameters | Value |
|---|---|
| batch size | 256 |
| buffer size | 25 000 |
| cost loss coefficient $\beta_c$ | 10 |
| discount factor $\gamma$ | 0.99 |
| feature freezing time $t_{\mathrm{freeze}}$ [environment steps] | 50 000 |
| learning rate $\alpha$ | 0.001 |
| reconstruction loss coefficient $\beta_g$ | 5 |
| reward loss coefficient $\beta_r$ | 0.25 |
| total time steps [environment steps] | 500 000 |
| training frequency [environment steps per training step] | 10 |
| training start time [environment steps] | 15 000 |
| target network frequency $k_{\mathrm{sync}}$ [environment steps per synchronization] | 500 |
| update steps [iterations per training step] | 10 |

| $\varepsilon$-**Annealing** | Value |
|---|---|
| end time [environment steps] | 100 000 |
| end value $\varepsilon_\infty$ | 0.25 |
| start time [environment steps] | 20 000 |
| start value $\varepsilon_0$ | 1 |

| Lagrangian Parameter Tuning | Value |
|---|---|
| cost budget $b$ | 5 |
| learning rate $\alpha_\lambda$ | 0.001 |
| start value $\lambda_0$ | 0 |
| training start time $t_{\mathrm{penalty}}$ [environment steps] | 100 000 |

| Feature Network $\phi$ | # Nodes |
|---|---|
| dense layer | 64 |
| ReLU activation function | |
| dense layer | 64 |
| ReLU activation function | |
| dense layer | 128 |
| normalization layer | |

| Inverse Network $g$ | # Nodes |
|---|---|
| dense layer | 128 |
| ReLU activation function | |
| dense layer | 64 |
| ReLU activation function | |
| dense layer | 64 |

| Successor Feature Network $M$ | # Nodes |
|---|---|
| dense layer | 128 |
| ReLU activation function | |
| dense layer | 128 |
| ReLU activation function | |
| dense layer | 128 |

## C Algorithm Sketches

### Successor Feature Network Update

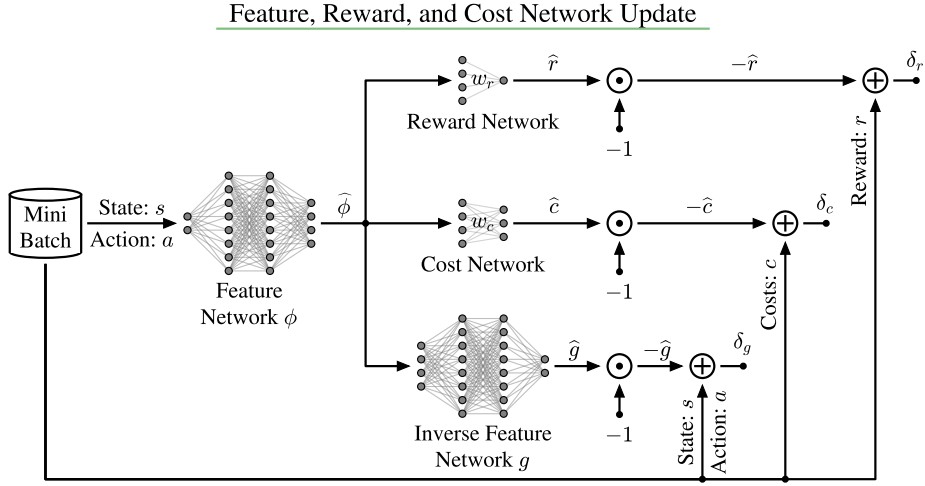

Figure 10: **Overview of the successor feature network update of the SafeDSR agent.** The agent samples mini batches $(s, a, s')$ uniformly from the replay buffer. The upper branch calculates the future expected state occupancy $\widehat{M}$ for state–action pairs $(s, a)$ in a high-dimensional space. The two lower branches estimate the target $\widehat{\phi}_{\mathrm{prev}} + \gamma \widehat{M}'_{\mathrm{prev}} := \phi_{\mathrm{prev}}(s, a) + \gamma M_{\mathrm{prev}}(\phi_{\mathrm{prev}}(s', a'))$, where $a'$ is the action the current network $M$ would select in $s'$. The target networks $M_{\mathrm{prev}}$ and $\phi_{\mathrm{prev}}$ are synchronized with the corresponding main networks every $k_{\mathrm{sync}}$ steps. The square of the temporal difference $\left\lVert \delta_M \right\rVert_2^2 = \left\lVert \widehat{\phi}_{\mathrm{prev}} + \gamma \widehat{M}'_{\mathrm{prev}} - \widehat{M} \right\rVert_2^2$ is used as the successor feature loss, which is optimized via gradient descent.

### Feature, Reward, and Cost Network Update

Figure 11: **Overview of the feature, reward, and cost network update of the SafeDSR agent.** The agent samples mini batches $(s, a, r, c)$ from the replay buffer using biased sampling. The upper branch calculates the reward difference $\delta_r$, the branch in the middle the cost difference $\delta_c$, and the lower branch the reconstruction difference $\delta_g$ for the state-action pairs $(s, a)$. All these differences are squared and summed up, yielding the feature loss $L_{\mathrm{features}} = \delta_r^2 + \lVert \delta_c \rVert_2^2 + \lVert \delta_g \rVert_2^2$, which is optimized via gradient descent.

# D    Environment Examples

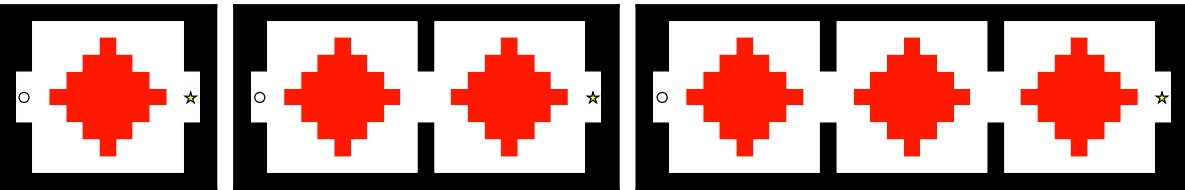

Figure 12: **Constrained grid environment with different number of rooms.** The exploration difficulty increases from left to right.

# E    Small Diamond to Large Diamond

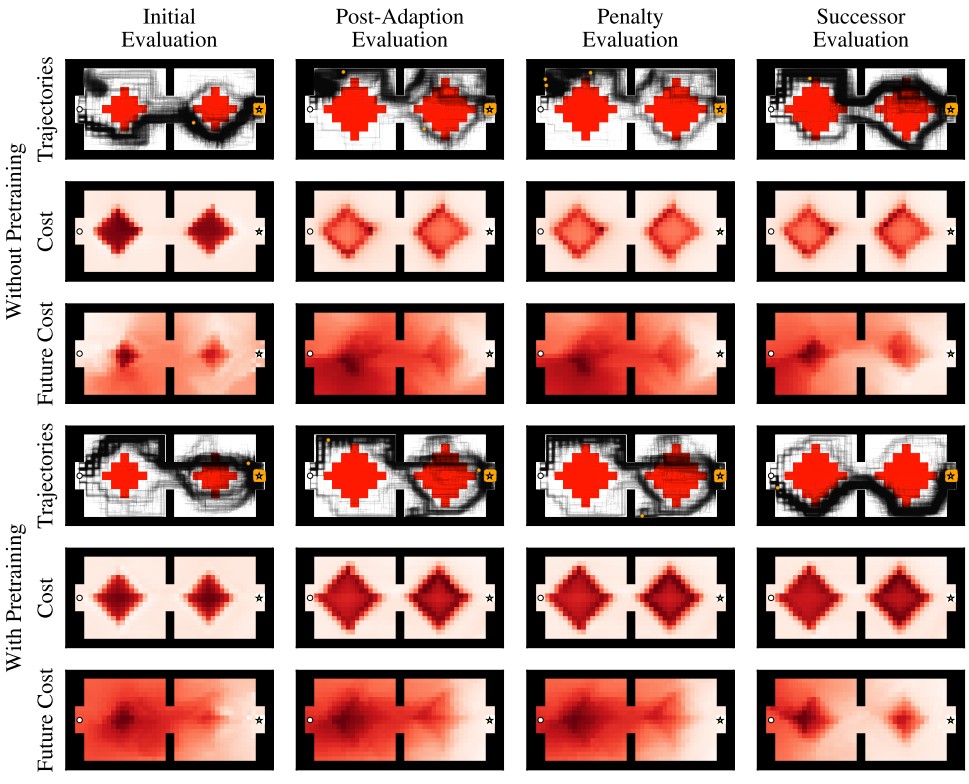

Figure 13: **Translation from a model pretrained on a smaller "diamond" shape towards a larger "diamond" distribution (Figure 3).** We refer to training reward and cost functions as "post-adaptation" evaluation, to training the Lagrangian on-policy as "penalty" evaluation, and finally to training the full network as "successor" evaluation. We also show the performance of our method with or without pretraining the feature embedding model.

## F   Baseline Comparison: Goal and Safe Goal Count

Table 3: **Goal count for the different baseline algorithms, SafeDQN, and SafeDSR in the environments without costs.** Note that the interquartile mean (IQM) is also reported in Figure 4.

| One Room | Mean | Std | 25th Percentile | IQM | 75th Percentile |
|---|---|---|---|---|---|
| CPO | 24 975.3 | 994.2 | 24 015.0 | 25 016.3 | 25 677.0 |
| CPPO-PID | 28 058.5 | 1161.6 | 27 394.0 | 28 130.5 | 29 048.8 |
| IPO | 28 984.7 | 718.5 | 28 289.0 | 28 957.7 | 29 414.5 |
| OnCRPO | 24 975.3 | 994.2 | 24 015.0 | 25 016.3 | 25 677.0 |
| P3O | 28 058.5 | 1161.6 | 27 394.0 | 28 130.5 | 29 048.8 |
| PCPO | 21 824.9 | 1381.3 | 20 777.8 | 21 917.3 | 22 936.0 |
| PDO | 24 326.4 | 10 630.4 | 20 847.5 | 27 272.2 | 32 081.0 |
| PPO-Lag | 28 058.5 | 1161.6 | 27 394.0 | 28 130.5 | 29 048.8 |
| RCPO | 24 975.3 | 994.2 | 24 015.0 | 25 016.3 | 25 677.0 |
| SafeDQN | 22 753.9 | 41.8 | 22 718.0 | 22 749.5 | 22 768.5 |
| SafeDSR | 21 181.7 | 2580.1 | 21 826.8 | 22 101.0 | 22 246.2 |
| TRPO-Lag | 24 975.3 | 994.2 | 24 015.0 | 25 016.3 | 25 677.0 |
| TRPO-PID | 24 975.3 | 994.2 | 24 015.0 | 25 016.3 | 25 677.0 |

| Two Rooms | Mean | Std | 25th Percentile | IQM | 75th Percentile |
|---|---|---|---|---|---|
| CPO | 10 317.1 | 3756.8 | 10 206.2 | 11 507.7 | 11 992.5 |
| CPPO-PID | 11 706.9 | 2369.5 | 10 581.8 | 12 277.8 | 13 163.0 |
| IPO | 12 489.0 | 2864.2 | 12 420.2 | 13 240.2 | 13 642.5 |
| OnCRPO | 10 317.1 | 3756.8 | 10 206.2 | 11 507.7 | 11 992.5 |
| P3O | 11 706.9 | 2369.5 | 10 581.8 | 12 277.8 | 13 163.0 |
| PCPO | 7743.6 | 4242.0 | 7340.8 | 9306.5 | 10 086.2 |
| PDO | 6915.7 | 5821.5 | 592.8 | 6919.5 | 11 460.8 |
| PPO-Lag | 11 706.9 | 2369.5 | 10 581.8 | 12 277.8 | 13 163.0 |
| RCPO | 10 311.5 | 3754.2 | 10 206.2 | 11 498.5 | 11 992.5 |
| SafeDQN | 11 334.6 | 76.7 | 11 261.0 | 11 340.0 | 11 403.0 |
| SafeDSR | 8661.3 | 3112.1 | 5781.2 | 9317.0 | 11 149.8 |
| TRPO-Lag | 10 317.1 | 3756.8 | 10 206.2 | 11 507.7 | 11 992.5 |
| TRPO-PID | 10 317.1 | 3756.8 | 10 206.2 | 11 507.7 | 11 992.5 |

| Three Rooms | Mean | Std | 25th Percentile | IQM | 75th Percentile |
|---|---|---|---|---|---|
| CPO | 4884.6 | 3357.0 | 1726.5 | 5523.0 | 7311.2 |
| CPPO-PID | 6568.3 | 2202.7 | 4466.8 | 6827.7 | 8234.5 |
| IPO | 5682.8 | 3697.1 | 3004.2 | 6553.7 | 8476.5 |
| OnCRPO | 4884.6 | 3357.0 | 1726.5 | 5523.0 | 7311.2 |
| P3O | 6568.3 | 2202.7 | 4466.8 | 6827.7 | 8234.5 |
| PCPO | 3850.1 | 3118.1 | 277.0 | 4160.3 | 6239.2 |
| PDO | 2915.0 | 3512.1 | 17.5 | 2187.2 | 6083.5 |
| PPO-Lag | 6568.3 | 2202.7 | 4466.8 | 6827.7 | 8234.5 |
| RCPO | 4884.5 | 3356.9 | 1726.5 | 5522.8 | 7311.0 |
| SafeDQN | 7502.9 | 70.4 | 7463.0 | 7518.3 | 7558.2 |
| SafeDSR | 4627.2 | 2443.2 | 3106.0 | 4701.7 | 7230.0 |
| TRPO-Lag | 4884.6 | 3357.0 | 1726.5 | 5523.0 | 7311.2 |
| TRPO-PID | 4884.6 | 3357.0 | 1726.5 | 5523.0 | 7311.2 |

Table 4: **Goal count for the different baseline algorithms, SafeDQN, and SafeDSR in the environments with costs.** Note that the interquartile mean (IQM) is also reported in Figure 4.

| One Room | Mean | Std | 25th Percentile | IQM | 75th Percentile |
|---|---|---|---|---|---|
| CPO | 7735.4 | 3611.2 | 7741.2 | 8818.3 | 9744.2 |
| CPPO-PID | 7307.8 | 4587.9 | 4844.8 | 6883.8 | 9879.2 |
| IPO | 9686.6 | 2342.8 | 7870.8 | 9828.0 | 11 338.2 |
| OnCRPO | 8696.6 | 4818.9 | 8235.5 | 10 255.5 | 11 755.2 |
| P3O | 525.9 | 758.5 | 45.8 | 251.5 | 745.2 |
| PCPO | 1641.0 | 2973.9 | 501.2 | 742.3 | 909.5 |
| PDO | 14 748.9 | 10 260.0 | 9191.2 | 15 193.2 | 24 075.2 |
| PPO-Lag | 25 727.1 | 2434.0 | 23 306.5 | 25 753.5 | 27 345.8 |
| RCPO | 23 271.9 | 1690.5 | 21 738.5 | 23 342.0 | 24 595.5 |
| SafeDQN | 15 933.0 | 587.5 | 15 739.2 | 15 966.3 | 16 098.8 |
| SafeDSR | 6632.4 | 695.0 | 6127.8 | 6713.5 | 7153.2 |
| TRPO-Lag | 23 271.9 | 1690.5 | 21 738.5 | 23 342.0 | 24 595.5 |
| TRPO-PID | 6868.3 | 4262.1 | 5527.2 | 7514.0 | 9831.0 |

| Two Rooms | Mean | Std | 25th Percentile | IQM | 75th Percentile |
|---|---|---|---|---|---|
| CPO | 42.5 | 36.6 | 7.8 | 40.2 | 77.2 |
| CPPO-PID | 23.1 | 22.0 | 10.5 | 19.2 | 29.0 |
| IPO | 120.4 | 74.2 | 69.2 | 113.7 | 165.2 |
| OnCRPO | 66.1 | 55.7 | 7.8 | 67.0 | 118.8 |
| P3O | 14.8 | 13.9 | 3.5 | 13.0 | 28.8 |
| PCPO | 184.7 | 188.1 | 64.8 | 139.8 | 230.5 |
| PDO | 441.6 | 608.3 | 139.0 | 259.8 | 374.8 |
| PPO-Lag | 8855.5 | 4070.7 | 6963.0 | 10 034.2 | 11 962.2 |
| RCPO | 8392.1 | 3374.5 | 7283.8 | 9172.2 | 10 283.8 |
| SafeDQN | 1756.2 | 1442.0 | 1053.0 | 1213.3 | 1633.2 |
| SafeDSR | 3534.0 | 382.6 | 3194.8 | 3490.7 | 3759.0 |
| TRPO-Lag | 8393.4 | 3375.4 | 7283.8 | 9172.2 | 10 283.8 |
| TRPO-PID | 20.1 | 19.9 | 11.8 | 15.0 | 19.2 |

| Three Rooms | Mean | Std | 25th Percentile | IQM | 75th Percentile |
|---|---|---|---|---|---|
| CPO | 7.0 | 8.8 | 1.2 | 3.8 | 6.5 |
| CPPO-PID | 7.2 | 10.1 | 1.0 | 4.5 | 10.2 |
| IPO | 19.0 | 23.3 | 1.0 | 14.2 | 37.0 |
| OnCRPO | 7.6 | 9.7 | 1.2 | 4.3 | 10.0 |
| P3O | 2.0 | 2.7 | 0.0 | 1.0 | 2.0 |
| PCPO | 63.3 | 88.5 | 4.2 | 32.5 | 76.2 |
| PDO | 31.6 | 38.8 | 0.2 | 21.7 | 56.0 |
| PPO-Lag | 2422.7 | 2300.2 | 139.2 | 2216.7 | 4030.5 |
| RCPO | 1879.6 | 2318.1 | 1.0 | 1454.7 | 4067.2 |
| SafeDQN | 588.1 | 61.4 | 571.5 | 594.7 | 634.2 |
| SafeDSR | 1246.8 | 629.8 | 1040.0 | 1324.0 | 1569.5 |
| TRPO-Lag | 1879.6 | 2318.1 | 1.0 | 1454.7 | 4067.2 |
| TRPO-PID | 5.4 | 5.8 | 1.2 | 3.7 | 5.8 |

Table 5: **Safe goal count for the different baseline algorithms, SafeDQN, and SafeDSR in the environments with costs.** Note that the interquartile mean (IQM) is also reported in Figure 4.

| One Room | Mean | Std | 25th Percentile | IQM | 75th Percentile |
|---|---|---|---|---|---|
| CPO | 2752.6 | 1172.7 | 2716.2 | 3056.5 | 3520.2 |
| CPPO-PID | 5434.8 | 2486.3 | 4511.8 | 5766.5 | 6732.8 |
| IPO | 3530.5 | 483.5 | 3232.0 | 3466.8 | 3952.2 |
| OnCRPO | 1547.0 | 979.2 | 1110.5 | 1689.8 | 2232.2 |
| P3O | 412.2 | 594.3 | 20.2 | 204.7 | 659.2 |
| PCPO | 350.8 | 1086.3 | 0.2 | 2.0 | 4.5 |
| PDO | 2052.3 | 2305.9 | 187.2 | 1614.7 | 4211.8 |
| PPO-Lag | 1093.5 | 910.2 | 256.8 | 1011.8 | 1877.0 |
| RCPO | 563.7 | 542.7 | 143.0 | 448.0 | 939.5 |
| SafeDQN | 8138.9 | 522.8 | 7826.0 | 8163.5 | 8474.8 |
| SafeDSR | 3806.9 | 259.5 | 3604.8 | 3791.7 | 3975.8 |
| TRPO-Lag | 563.7 | 542.7 | 143.0 | 448.0 | 939.5 |
| TRPO-PID | 6034.6 | 3477.8 | 5370.8 | 6898.5 | 8439.0 |

| Two Rooms | Mean | Std | 25th Percentile | IQM | 75th Percentile |
|---|---|---|---|---|---|
| CPO | 4.1 | 5.2 | 0.0 | 2.8 | 8.5 |
| CPPO-PID | 0.4 | 1.0 | 0.0 | 0.0 | 0.0 |
| IPO | 12.4 | 13.6 | 4.0 | 9.0 | 24.0 |
| OnCRPO | 5.7 | 6.8 | 0.0 | 4.3 | 9.5 |
| P3O | 1.1 | 1.4 | 0.0 | 0.7 | 1.8 |
| PCPO | 0.0 | 0.0 | 0.0 | 0.0 | 0.0 |
| PDO | 5.7 | 8.4 | 0.5 | 3.0 | 7.0 |
| PPO-Lag | 5.2 | 5.7 | 2.2 | 3.5 | 4.8 |
| RCPO | 3.9 | 2.9 | 2.0 | 3.7 | 6.2 |
| SafeDQN | 469.0 | 964.6 | 15.2 | 97.5 | 361.0 |
| SafeDSR | 2147.0 | 213.7 | 1970.8 | 2137.5 | 2324.8 |
| TRPO-Lag | 3.9 | 2.9 | 2.0 | 3.7 | 6.2 |
| TRPO-PID | 0.3 | 0.9 | 0.0 | 0.0 | 0.0 |

| Three Rooms | Mean | Std | 25th Percentile | IQM | 75th Percentile |
|---|---|---|---|---|---|
| CPO | 0.0 | 0.0 | 0.0 | 0.0 | 0.0 |
| CPPO-PID | 0.0 | 0.0 | 0.0 | 0.0 | 0.0 |
| IPO | 0.0 | 0.0 | 0.0 | 0.0 | 0.0 |
| OnCRPO | 0.0 | 0.0 | 0.0 | 0.0 | 0.0 |
| P3O | 0.0 | 0.0 | 0.0 | 0.0 | 0.0 |
| PCPO | 0.0 | 0.0 | 0.0 | 0.0 | 0.0 |
| PDO | 0.0 | 0.0 | 0.0 | 0.0 | 0.0 |
| PPO-Lag | 0.1 | 0.3 | 0.0 | 0.0 | 0.0 |
| RCPO | 0.1 | 0.3 | 0.0 | 0.0 | 0.0 |
| SafeDQN | 2.8 | 6.9 | 0.0 | 0.2 | 0.8 |
| SafeDSR | 645.2 | 430.3 | 351.5 | 681.8 | 900.8 |
| TRPO-Lag | 0.1 | 0.3 | 0.0 | 0.0 | 0.0 |
| TRPO-PID | 0.0 | 0.0 | 0.0 | 0.0 | 0.0 |

## G   Baseline Comparison: Average Episodic Reward and Cost

Table 6: **Average episodic reward for the different baseline algorithms, SafeDQN, and SafeDSR in the environments with costs** at the end of training. Note that the interquartile mean (IQM) is also reported in Figure 5.

| One Room | Mean | Std | 25th Percentile | IQM | 75th Percentile |
|---|---|---|---|---|---|
| CPO | $-0.4$ | 3.4 | 0.7 | 0.7 | 0.8 |
| CPPO-PID | 0.7 | 0.2 | 0.7 | 0.7 | 0.8 |
| IPO | 0.8 | 0.0 | 0.7 | 0.8 | 0.8 |
| OnCRPO | $-1.4$ | 4.6 | 0.7 | 0.8 | 0.8 |
| P3O | $-5.3$ | 4.8 | $-10.0$ | $-5.6$ | $-0.8$ |
| PCPO | $-5.7$ | 4.1 | $-9.7$ | $-6.1$ | $-2.0$ |
| PDO | $-1.3$ | 4.6 | 0.8 | 0.8 | 0.9 |
| PPO-Lag | 0.9 | 0.0 | 0.9 | 0.9 | 0.9 |
| RCPO | 0.9 | 0.0 | 0.9 | 0.9 | 0.9 |
| SafeDQN | 0.7 | 0.1 | 0.7 | 0.7 | 0.8 |
| SafeDSR | 0.3 | 0.2 | 0.2 | 0.3 | 0.4 |
| TRPO-Lag | 0.9 | 0.0 | 0.9 | 0.9 | 0.9 |
| TRPO-PID | $-1.4$ | 4.5 | 0.7 | 0.7 | 0.8 |

| Two Rooms | Mean | Std | 25th Percentile | IQM | 75th Percentile |
|---|---|---|---|---|---|
| CPO | $-10.0$ | 0.0 | $-10.0$ | $-10.0$ | $-10.0$ |
| CPPO-PID | $-10.0$ | 0.0 | $-10.0$ | $-10.0$ | $-10.0$ |
| IPO | $-6.5$ | 2.7 | $-9.2$ | $-6.4$ | $-4.5$ |
| OnCRPO | $-9.4$ | 2.0 | $-10.0$ | $-10.0$ | $-10.0$ |
| P3O | $-9.8$ | 0.5 | $-10.0$ | $-10.0$ | $-10.0$ |
| PCPO | $-7.4$ | 3.3 | $-10.0$ | $-8.1$ | $-4.8$ |
| PDO | $-8.7$ | 2.9 | $-10.0$ | $-10.0$ | $-10.0$ |
| PPO-Lag | 0.5 | 0.6 | 0.6 | 0.7 | 0.7 |
| RCPO | $-0.4$ | 3.4 | 0.7 | 0.7 | 0.7 |
| SafeDQN | $-4.9$ | 5.0 | $-10.0$ | $-4.9$ | 0.1 |
| SafeDSR | 0.1 | 0.3 | 0.1 | 0.2 | 0.3 |
| TRPO-Lag | $-0.4$ | 3.4 | 0.7 | 0.7 | 0.7 |
| TRPO-PID | $-10.0$ | 0.0 | $-10.0$ | $-10.0$ | $-10.0$ |

| Three Rooms | Mean | Std | 25th Percentile | IQM | 75th Percentile |
|---|---|---|---|---|---|
| CPO | $-10.0$ | 0.0 | $-10.0$ | $-10.0$ | $-10.0$ |
| CPPO-PID | $-10.0$ | 0.0 | $-10.0$ | $-10.0$ | $-10.0$ |
| IPO | $-9.3$ | 2.3 | $-10.0$ | $-10.0$ | $-10.0$ |
| OnCRPO | $-10.0$ | 0.0 | $-10.0$ | $-10.0$ | $-10.0$ |
| P3O | $-10.0$ | 0.0 | $-10.0$ | $-10.0$ | $-10.0$ |
| PCPO | $-7.8$ | 2.8 | $-10.0$ | $-8.4$ | $-5.1$ |
| PDO | $-10.0$ | 0.0 | $-10.0$ | $-10.0$ | $-10.0$ |
| PPO-Lag | $-2.5$ | 4.4 | $-4.5$ | $-1.0$ | 0.5 |
| RCPO | $-4.9$ | 5.4 | $-10.0$ | $-4.9$ | 0.5 |
| SafeDQN | $-10.0$ | 0.0 | $-10.0$ | $-10.0$ | $-10.0$ |
| SafeDSR | $-3.7$ | 4.4 | $-8.0$ | $-2.7$ | $-0.4$ |
| TRPO-Lag | $-4.9$ | 5.4 | $-10.0$ | $-4.9$ | 0.5 |
| TRPO-PID | $-10.0$ | 0.0 | $-10.0$ | $-10.0$ | $-10.0$ |

Table 7: **Average episodic cost for the different baseline algorithms, SafeDQN, and SafeDSR in the environments with costs** at the end of training. Note that the interquartile mean (IQM) is also reported in Figure 5.

| One Room | Mean | Std | 25th Percentile | IQM | 75th Percentile |
|---|---|---|---|---|---|
| CPO | 6.4 | 5.8 | 2.0 | 6.0 | 11.5 |
| CPPO-PID | 5.2 | 5.2 | 0.5 | 4.7 | 10.0 |
| IPO | 6.4 | 4.3 | 3.0 | 7.0 | 9.5 |
| OnCRPO | 9.2 | 6.9 | 3.5 | 9.3 | 13.5 |
| P3O | 0.2 | 0.6 | 0.0 | 0.0 | 0.0 |
| PCPO | 319.8 | 228.5 | 119.0 | 308.3 | 513.0 |
| PDO | 11.0 | 8.7 | 2.0 | 11.7 | 19.5 |
| PPO-Lag | 18.4 | 0.8 | 18.0 | 18.0 | 18.0 |
| RCPO | 18.6 | 1.0 | 18.0 | 18.3 | 19.5 |
| SafeDQN | 5.0 | 3.2 | 3.2 | 5.2 | 7.8 |
| SafeDSR | 5.4 | 5.8 | 1.0 | 4.0 | 8.8 |
| TRPO-Lag | 18.6 | 1.0 | 18.0 | 18.3 | 19.5 |
| TRPO-PID | 1.4 | 2.7 | 0.0 | 0.3 | 1.5 |

| Two Rooms | Mean | Std | 25th Percentile | IQM | 75th Percentile |
|---|---|---|---|---|---|
| CPO | 5.2 | 8.6 | 0.0 | 2.3 | 8.5 |
| CPPO-PID | 1.4 | 3.1 | 0.0 | 0.3 | 1.5 |
| IPO | 17.8 | 15.0 | 11.0 | 16.7 | 24.0 |
| OnCRPO | 4.0 | 10.0 | 0.0 | 0.3 | 1.5 |
| P3O | 4.4 | 9.3 | 0.0 | 0.0 | 0.0 |
| PCPO | 364.0 | 145.7 | 265.0 | 330.0 | 447.5 |
| PDO | 4.2 | 5.8 | 0.0 | 2.3 | 5.5 |
| PPO-Lag | 35.4 | 5.3 | 34.5 | 36.7 | 38.0 |
| RCPO | 31.8 | 11.9 | 32.5 | 35.7 | 38.0 |
| SafeDQN | 9.0 | 18.0 | 0.2 | 3.2 | 6.2 |
| SafeDSR | 3.5 | 3.5 | 1.0 | 2.8 | 5.5 |
| TRPO-Lag | 31.6 | 11.7 | 32.5 | 35.7 | 38.0 |
| TRPO-PID | 0.0 | 0.0 | 0.0 | 0.0 | 0.0 |

| Three Rooms | Mean | Std | 25th Percentile | IQM | 75th Percentile |
|---|---|---|---|---|---|
| CPO | 21.4 | 49.3 | 0.0 | 1.3 | 4.0 |
| CPPO-PID | 0.4 | 0.8 | 0.0 | 0.0 | 0.0 |
| IPO | 5.8 | 9.2 | 0.0 | 2.7 | 11.0 |
| OnCRPO | 16.0 | 40.4 | 0.0 | 2.3 | 7.0 |
| P3O | 0.2 | 0.6 | 0.0 | 0.0 | 0.0 |
| PCPO | 396.4 | 141.1 | 299.5 | 383.7 | 467.0 |
| PDO | 4.0 | 6.0 | 0.0 | 2.0 | 7.0 |
| PPO-Lag | 33.8 | 18.1 | 29.5 | 39.3 | 47.0 |
| RCPO | 23.8 | 25.6 | 0.0 | 21.3 | 49.5 |
| SafeDQN | 7.9 | 9.7 | 0.0 | 5.3 | 13.8 |
| SafeDSR | 26.0 | 63.5 | 1.5 | 5.8 | 10.5 |
| TRPO-Lag | 23.8 | 25.6 | 0.0 | 21.3 | 49.5 |
| TRPO-PID | 0.2 | 0.6 | 0.0 | 0.0 | 0.0 |

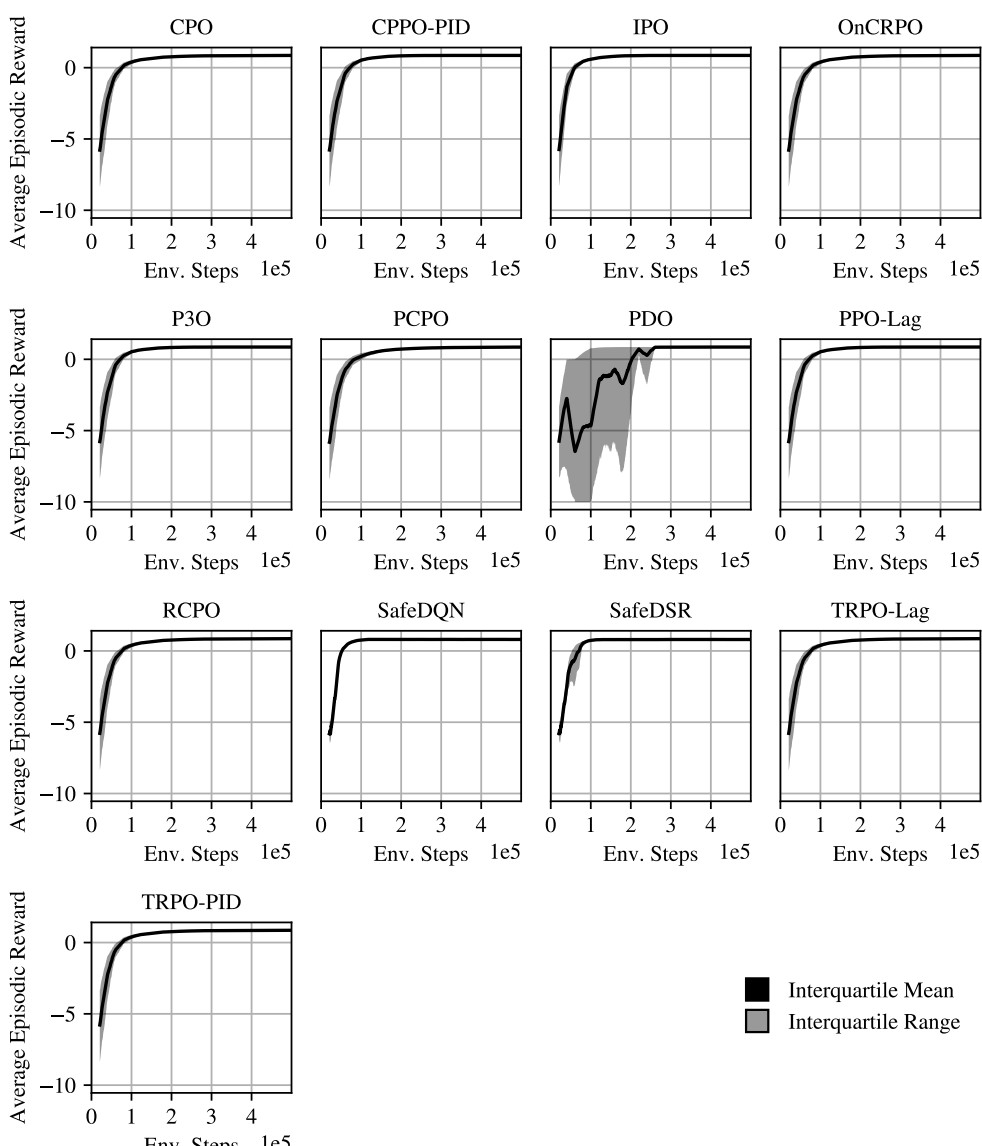

Figure 14: **Average episodic reward for the different baseline algorithms, SafeDQN, and SafeDSR in the one-room environment without costs.** The window average is calculated as discussed in Section 5.2.

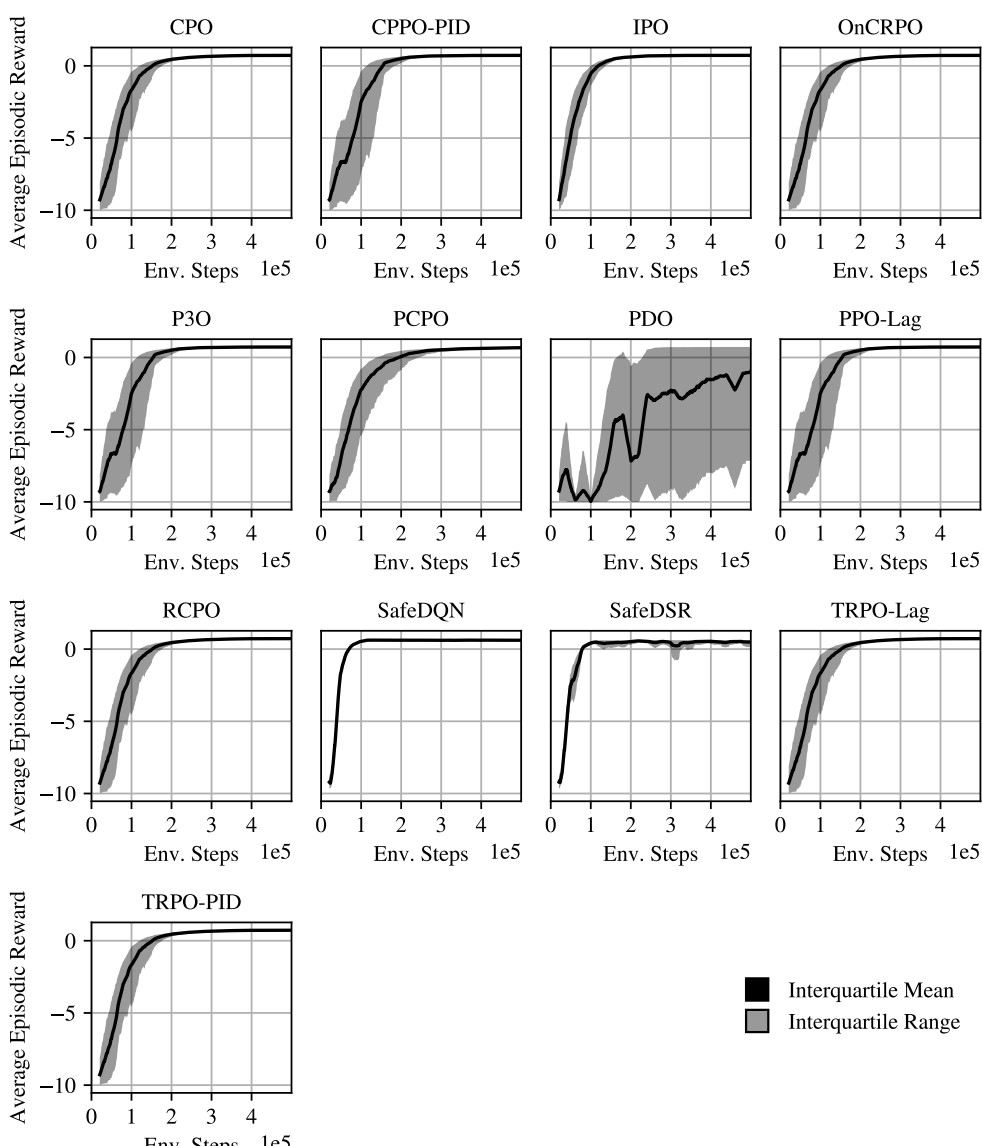

Figure 15: **Average episodic reward for the different baseline algorithms, SafeDQN, and SafeDSR in the two-room environment without costs.** The window average is calculated as discussed in Section 5.2.

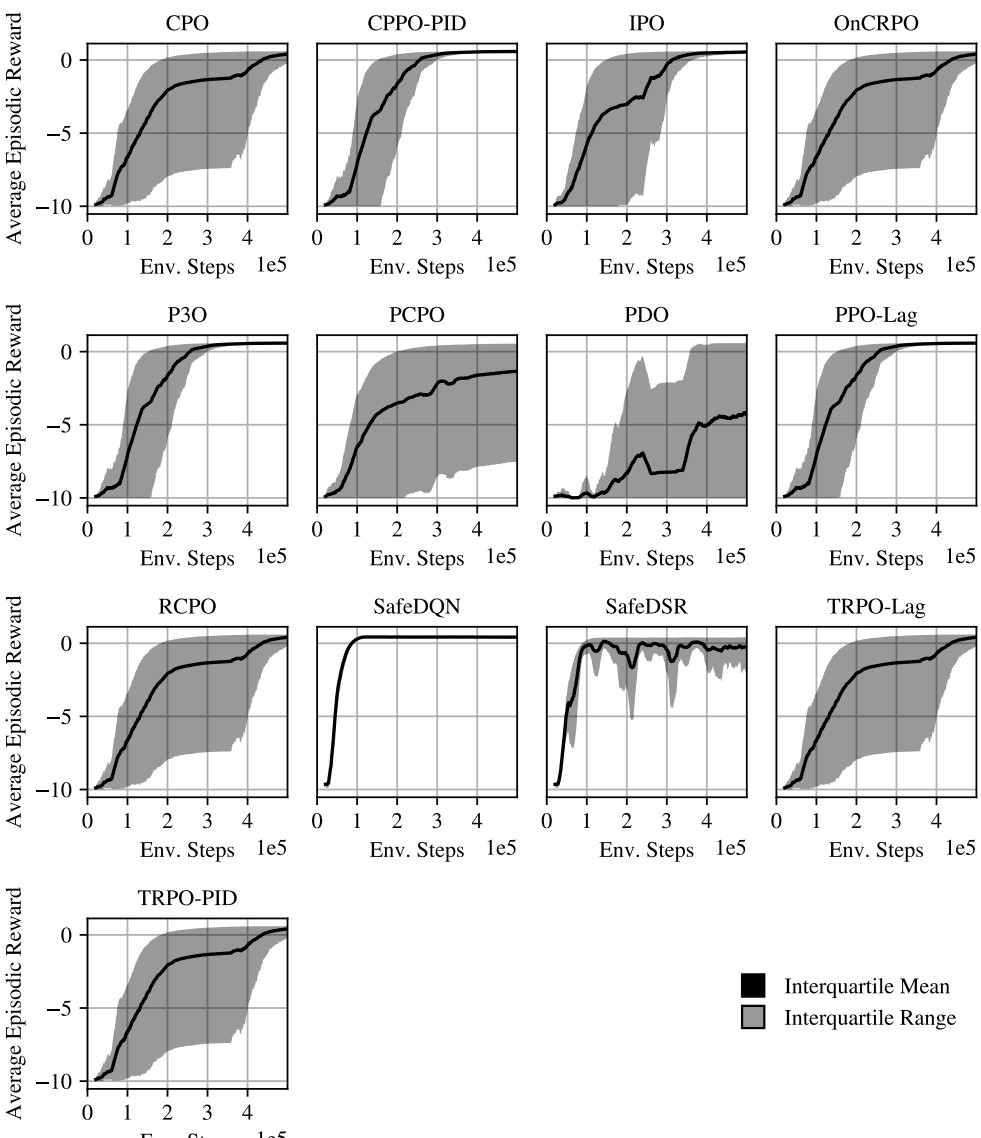

Figure 16: **Average episodic reward for the different baseline algorithms, SafeDQN, and SafeDSR in the three-room environment without costs.** The window average is calculated as discussed in Section 5.2.

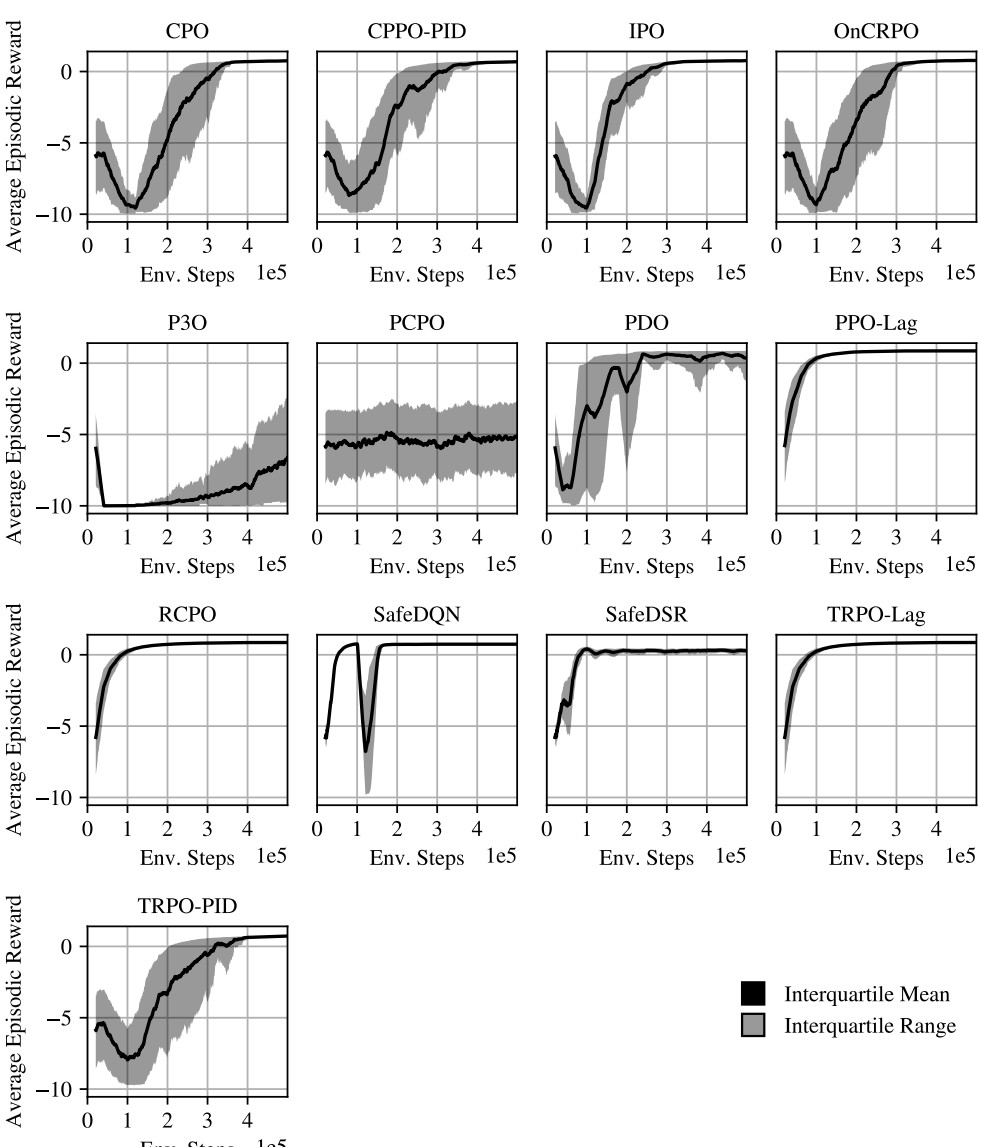

Figure 17: **Average episodic reward for the different baseline algorithms, SafeDQN, and SafeDSR in the one-room environment with costs.** The window average is calculated as discussed in Section 5.2.

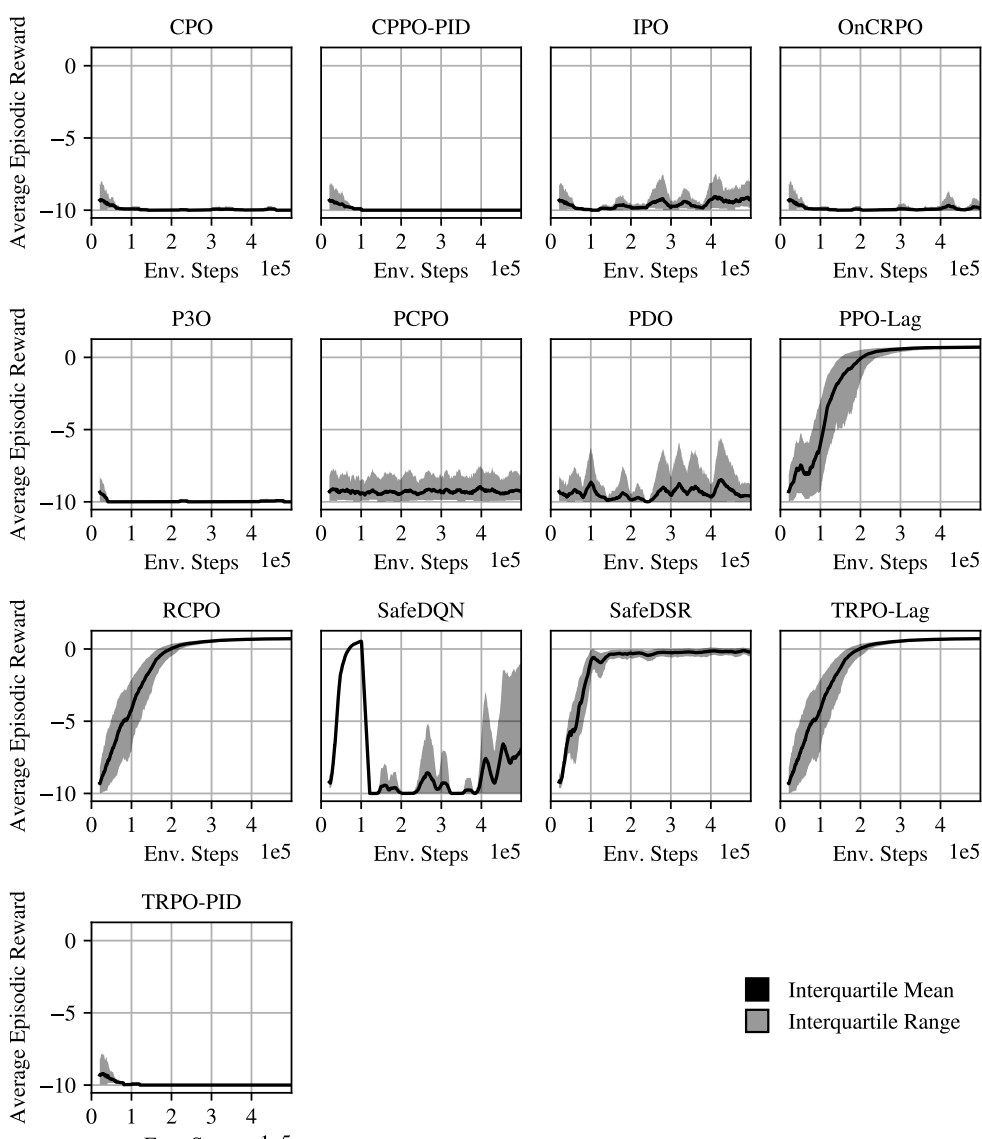

Figure 18: **Average episodic reward for the different baseline algorithms, SafeDQN, and SafeDSR in the two-room environment with costs.** The window average is calculated as discussed in Section 5.2.

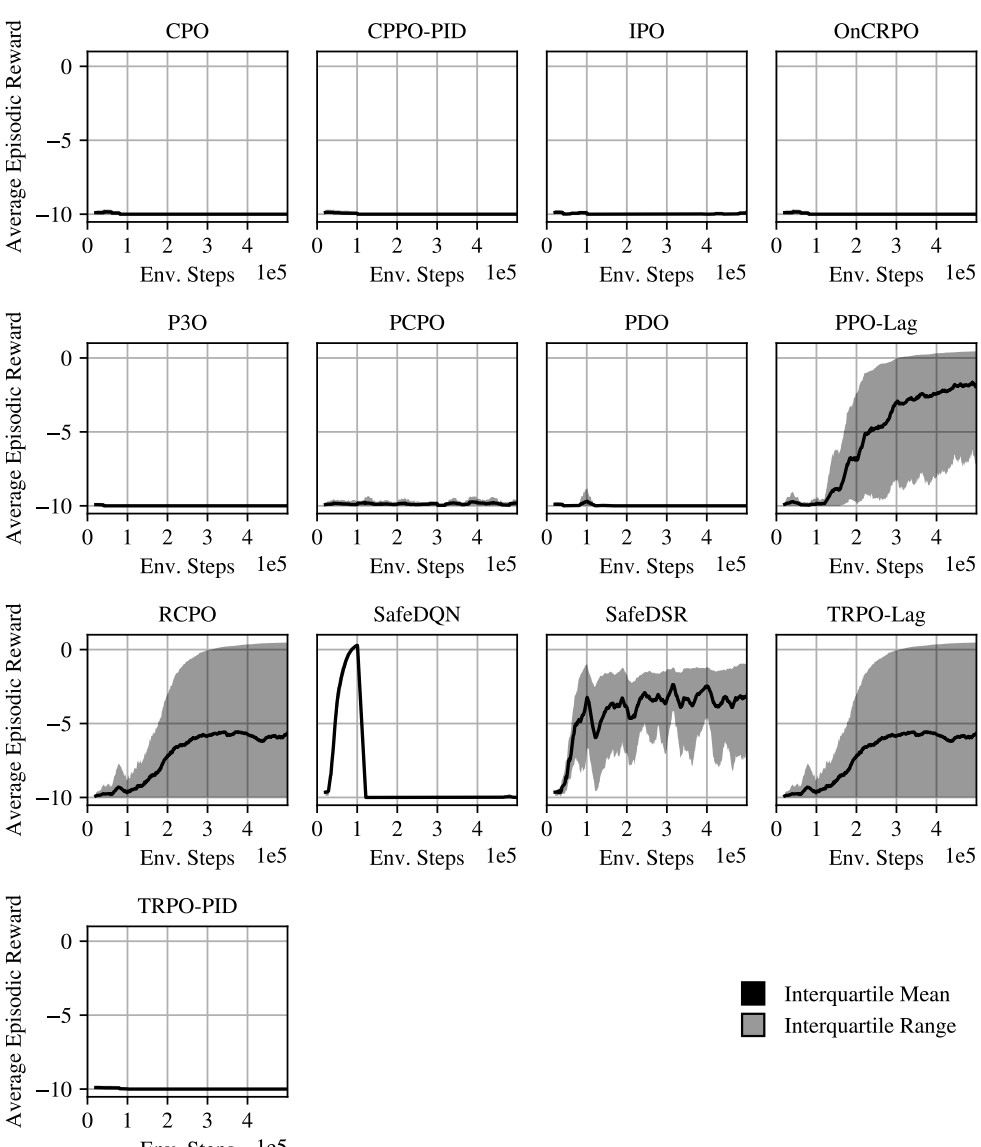

Figure 19: **Average episodic reward for the different baseline algorithms, SafeDQN, and SafeDSR in the three-room environment with costs.** The window average is calculated as discussed in Section 5.2.

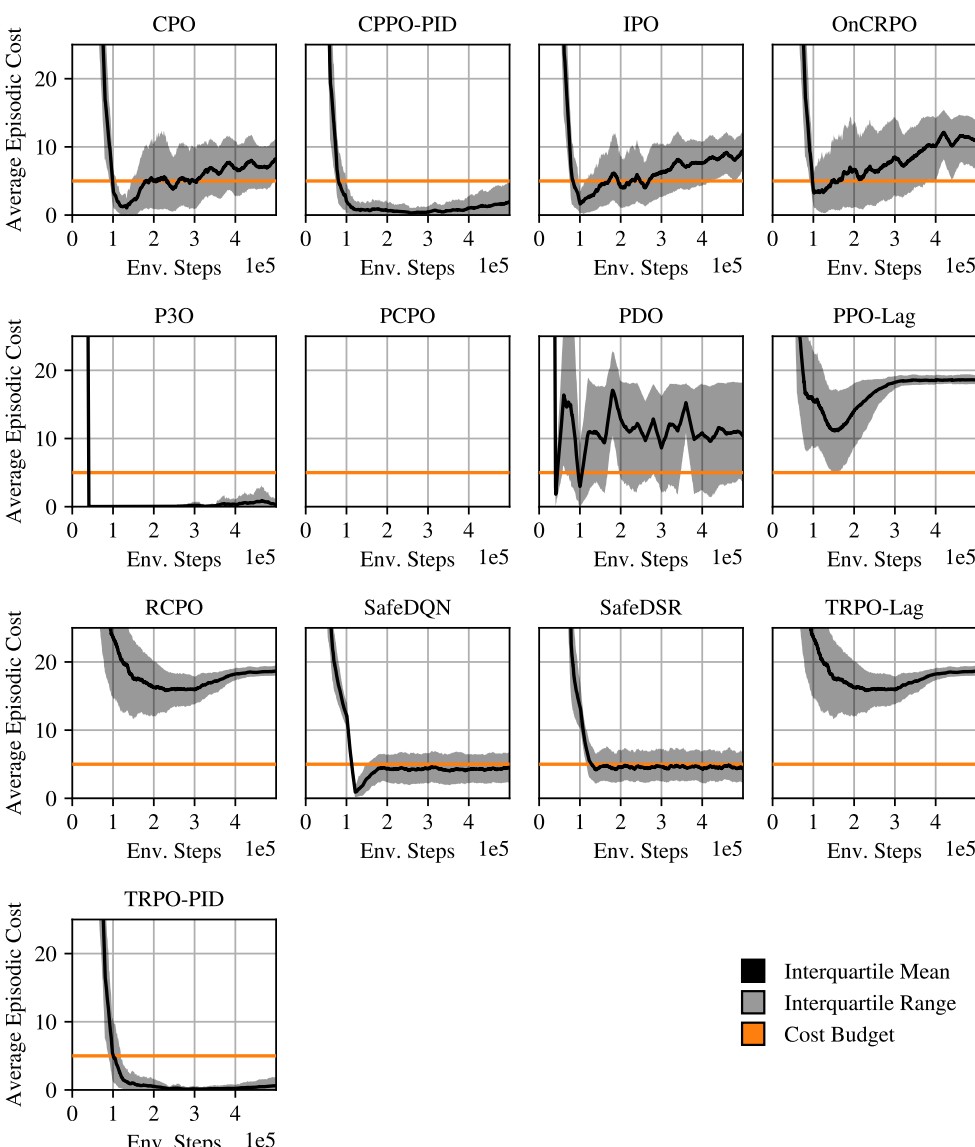

Figure 20: **Average episodic cost for the different baseline algorithms, SafeDQN, and SafeDSR in the one-room environment with costs.** The window average is calculated as discussed in Section 5.2. The average episodic cost of PCPO is too large for clear visibility. However, we retained uniform scaling across all plots for comparison purposes.

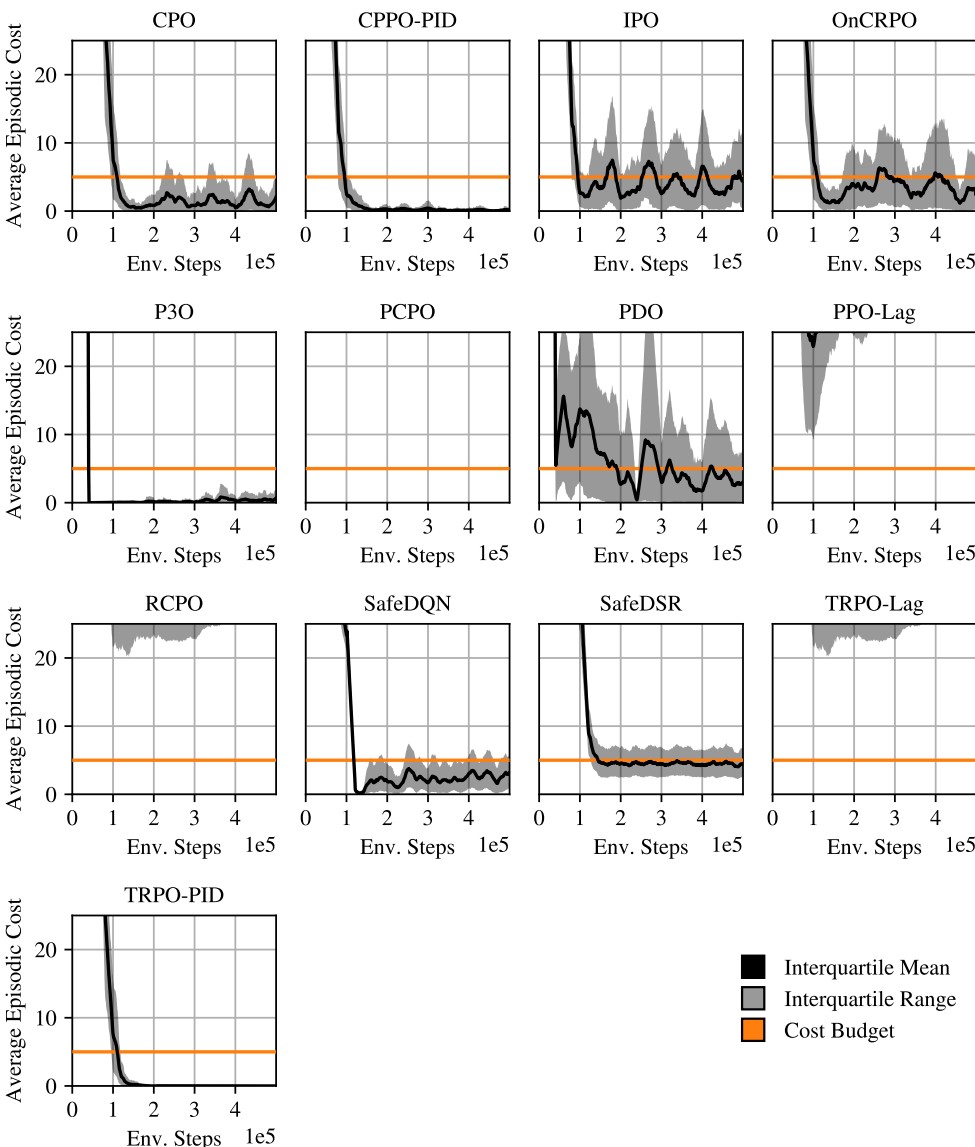

Figure 21: **Average episodic cost for the different baseline algorithms, SafeDQN, and SafeDSR in the two-room environment with costs.** The window average is calculated as discussed in Section 5.2. The average episodic cost of PCPO, PPO-Lag, RCPO, and TRPO-Lag is too large for clear visibility. However, we retained uniform scaling across all plots for comparison purposes.

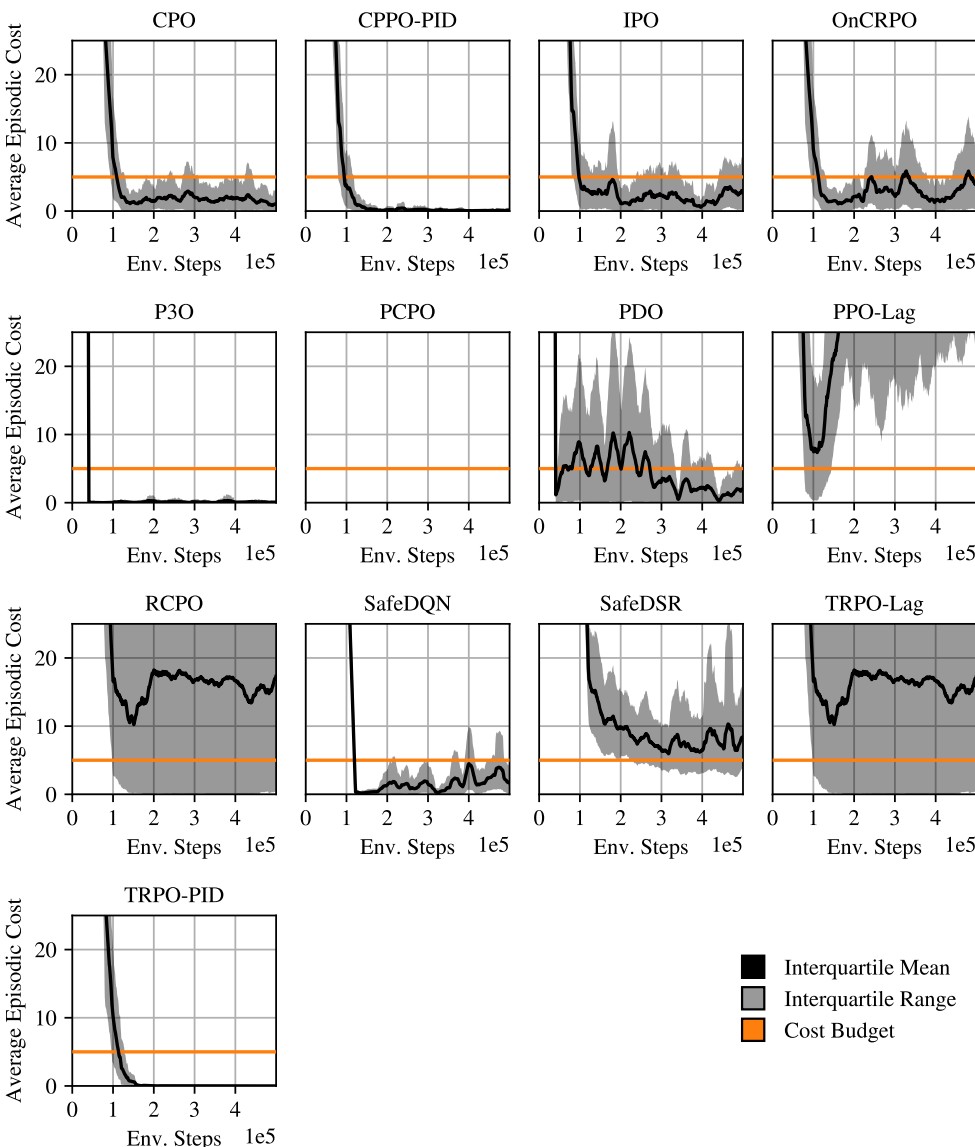

Figure 22: **Average episodic cost for the different baseline algorithms, SafeDQN, and SafeDSR in the three-room environment with costs.** The window average is calculated as discussed in Section 5.2. The average episodic cost of PCPO and PPO-Lag is too large for clear visibility. However, we retained uniform scaling across all plots for comparison purposes.

## H  Average Safe Goal Rate When Changing Cost Distributions

We showcase in Figure 23 the change in safe goal rate as the costs get adapted. The numbers correspond to the different phases outlined in Table 8. It is worth noting that the absolute performance depends strongly on the difficulty of the constraints. To give an extreme example: if the red constraints block the entire path, then the maximally achievable safe goal rate is zero. Since the safe area changes in all evaluations (e.g., "square" only has a single feasible line at the top and bottom), one cannot directly compare safe goal rates.

For this experiment, we used a simpler Lagrangian parameter update rule than shown in Equation (16). We increased the Lagrangian parameter by 0.01 if the current episode violates the corresponding constraint, and otherwise decreases it by 0.01. This has the effect, that the average safe goal rate during the first phase in Figure 23 is smaller than in Figure 6 (middle). However, this change has no influence on the qualitative analysis in Section 5.4.

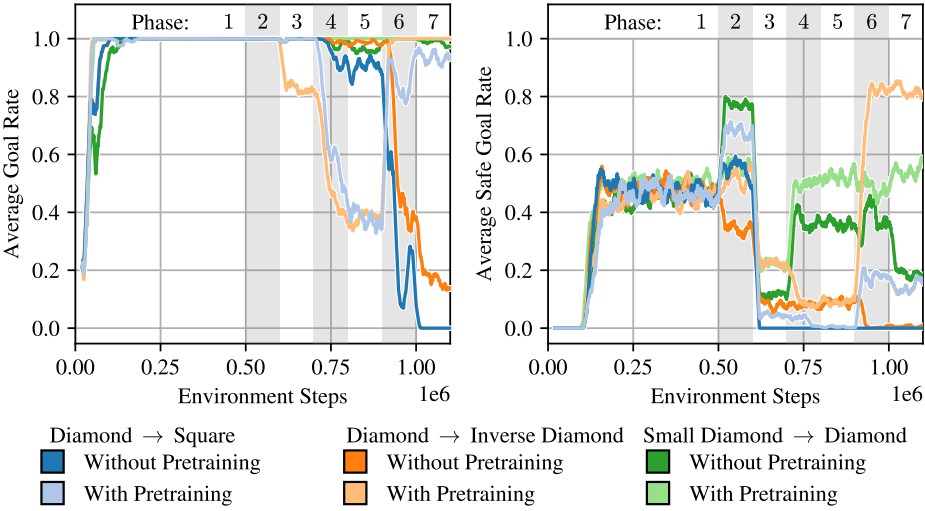

Figure 23: **Safe goal rate as the optimization cycles through different settings.** One has to be careful when comparing the different models against each other, as the achievable safe goal rate is highly dependent on the difficulty of the constraints.

| Phase | Name | Start Time |
|---|---|---|
| 1 | initial training | 0 |
| 2 | initial evaluation | 500 000 |
| 2.5 | environment adaption | 600 000 |
| 3 | post-adaption evaluation | 600 000 |
| 4 | penalty learning | 700 000 |
| 5 | penalty evaluation | 800 000 |
| 6 | successor learning | 900 000 |
| 7 | successor evaluation | 1 000 000 |

Table 8: **Different training and evaluation phases** of the experiment with changing cost distributions. We change the environment during the *environment adaption* phase, which lies between phase 2 and 3. The start time is given in environment steps and during evaluation we freeze all networks and parameters.

