# OpenReview forum: "Constrained Reinforcement Learning Using Successor Representations"
_TMLR — Accepted by TMLR_

### Review · Reviewer_peX1 · 2026-02-28

**Summary Of Contributions:**

This paper proposes a safe RL algorithms that decouples the environment dynamics, rewards and costs, so that the agent can adapt to new constraints and changed environments without training from scratch. The proposed approach is evaluated on a set of benchmark tasks, and the results show that it outperforms several baseline algorithms in terms of safety and performance. The authors also empirically demonstrate how their approach efficiently adapts to new constraints and changed environments.

**Audience:**

Yes

**Audience Explanation:**

The problem of efficiently adapting to new constraints and changed environments in safe RL has been a topic of significant interest. This work proposes some empirical contributions in this direction.

**Claims And Evidence:**

No

**Claims Explanation:**

Although this work provides some useful empirical results, there are some issues that remains to be addressed for a better understanding of the proposed approach and its limitations.

* According to, e.g., figure 6, it seems that the trajectories generated by SafeDQN and SafeDSR still visit some unsafe states. The question is then: is the approach proposed in this work able to handle hard constraints, e.g., some states are not allowed to be visited?
* Following up on the previous question, the authors mentioned that random observations are required during the 'pretraining' phase. How does the pretraining work if it is not allowed (e.g., due to safety constraints) to visit some states during the pretraining phase? Is there a way to incorporate such constraints into the pretraining phase?
* In the loss function formulations, e.g., (14), all the terms are weighted equally. Is it possible to use different weights for different terms? How would that affect the performance of the algorithm? Additionally, do these loss terms have the same scale? If not, how to ensure that they are balanced during training?
* In the second paragraph below figure 5, the authors mention that "many of the tested algorithms are limited by their exploration strategies". Can the authors provide more details on this? Specifically, if the benchmark algorithms are set to use the same exploration strategy as SafeDQN, would that improve their performance?
* The experiments presented in this manuscript are limited in scale. The authors mentioned that this is due to interpretability concerns. However, it would be beneficial to see how the proposed approach performs on larger and more complex environments, such as those with higher-dimensional state spaces or more complex dynamics. This would help to better understand the scalability and generalization capabilities of the proposed approach.
* The discussion of recent works is limited, and the baselines are also a bit outdated. The authors should at least discuss more recent works in the related work section, and ideally, they should also include some of these recent works as baselines in the experiments. This would help to better position the proposed approach in the context of the current state-of-the-art in safe reinforcement learning.

**Requested Changes:**

The authors should address the questions listed in the above sections, and also consider fixing the following minor issues.

* Inconsistent use of "arg max" and "argmax". For example, the first form appears in Figure 2, while the second form appears in (15). The authors should choose one form and use it consistently throughout the manuscript.
* It is generally not considered as standard to refer to some equation before it is introduced, e.g., the update rule (16) is mentioned before it is defined. The authors should consider reordering the content to avoid this issue.

---

> ### Author Response · Authors · 2026-03-20
>
> Thank you for your reviewing our work: We agree that we can clarify some of the limitations of our approach.
> Going through your concerns specifically:
> >According to, e.g., figure 6, it seems that the trajectories generated by SafeDQN and SafeDSR still visit some unsafe states.
>
> There are two parts to this question: First, the direct answer for SafeDQN and SafeDSR is that in our paper we still allow for a “violation budget” of 5, which means that the optimal policy is allowed to “collect” up to 5 steps on the red squares.
> This is a core assumption of CMDPs: If you are not allowed to violate the constraints at all (e.g. a budget of 0) and a deterministic optimal policy exists, the CMDP objective degenerates to a termination criterion and a penalty. Specifically, assuming the return is bounded by $R$ (which follows either from a finite time horizon or a $\gamma<1$), then a CMDP with $\mathbb{E}[C(\pi)] = 0$ corresponds to a policy trained on the environment with termination criterion $\text{immediate costs}\neq0$ and a termination penalty of $P>R$. Under those circumstances, the optimal policy is the same for both cases.
>
> CMDPs become interesting exactly when the violation depends on the _entire trajectory_ rather than the immediate cost of a state. In this setting, CMDPs are generally much harder to solve since they encode a multi-objective lagrangian problem which is implicitly sparse (one only gets a “violated” signal into the policy iff the costs are above the threshold). CMDPs are also usually harder than simple state-wise constraints since state constraints directly tell the algorithm “do not visit this state” compared to CMDPs where a state may or may not be visited depending on the dynamics induced by the policy.
>
> This leads us to the second part of your question:
> No, CMDPs cannot model hard constraints $C(a,s)\leq d$ unless the environment is very nice: Finite dimensional, deterministic (or at least bounded uncertainty set for actions), etc…
> If you don’t have those, then there is no way to guarantee a specific behavior is even observable during training. If you have unobserved states, their policy will not necessarily uphold the constraint, hencewhy CMDPs study expectations, rather than hard constraints. Modelling hard constraints is only possible if you know more about the environment than what one generally assumes of a CMDP. If you know nothing, there is no way to establish “safe” subsets of the environment where costs are below threshold.
>
> >Following up on the previous question, the authors mentioned that random observations are required during the 'pretraining' phase. How does the pretraining work if it is not allowed (e.g., due to safety constraints) to visit some states during the pretraining phase?
>
> This is one of the limitations of any “successor feature” based representation: At some point, one needs to be able to guarantee that the reward is representable as the linear regressor over the features. For benchmarking problems this is generally an issue (see also our comment to 5FbN), since those are not designed to propose such features. In “real world” problems this often ends up being easier since one knows more about the problem (e.g. by using embedding models or by encoding coordinates directly using RBF kernels).
>
> For our work, we have to additionally guarantee that the features in the “violation region” are trained at all, since otherwise the transfer will not be possible as the model might not be able to distinguish different unsafe states (after all, these were undersampled by design). We don’t believe this would be a major limitation in practice since one can choose a good feature representation directly. If that is impossible, one either
> -	has states which are always considered unsafe, in which case we don’t need to distinguish them
> -	has states which are sometimes safe, in which case one can gather data during the times where it is safe for pretraining

---

> > ### Author Response · Authors · 2026-03-20
> >
> > > Is it possible to use different weights for different terms?
> >
> > We actually did weight these components: they are already set in our hyperparameter table in the appendix, but are somehow not displayed in (14). We will fix this mistake. In general, we did not find the objective to be too sensitive to weights and simply set the coefficients such that the overall scale of the terms was about the same at the beginning of training.
> >
> > >In the second paragraph below figure 5, the authors mention that "many of the tested algorithms are limited by their exploration strategies".
> >
> > This is more a general observation of all CMDP solvers: Exploration inside a CMDP is always hard simply due to the tension between being safe (which is naturally hostile towards trying new things) and exploring new states (which naturally will incur higher costs than staying in the established space). SafeDQN and SafeDSR use simple epsilon-greedy search and appeared to have less issues finding the goal. This is not an advantage we claim for either of these methods, but rather a more general observation on other algorithms.
> >
> > >The experiments presented in this manuscript are limited in scale.
> >
> > The core issue with applying these algorithms in normal “black box” benchmarks is that those do not guarantee the linearizability of the reward necessary for successor representations. The actual problem we look at is nontrivial for RL to solve due to the long-horizon needed to solve the problem (which can be seen by the other SOTA CMDP solvers failing this task).
> > Generally RL benchmarks aim to have a maximally compact state space, while successor representations want a maximally descriptive (but not necessarily compact) state space.
> >
> > >The discussion of recent works is limited, and the baselines are also a bit outdated.
> >
> > We focused our discussion of recent work on the state-of-the-art literature in model-free CMDP solving. Specifically, our discussion is limited to the benchmarks in “omnisafe” which is a “stable-baselines” like implementation of CMDP algorithms. Are you thinking of any particular algorithm you want us to mention?
> > We specifically want to note here that safe reinforcement learning and constrained reinforcement learning, while related, are not the same problem: There are reasons to use constraints beyond just safety, and there are ways of improving safety without needing to model constraints.
> > Within the familiy of constrained RL, there are generally a lot of algorithms to solve constrained problems depending on what they exactly constrain: Constrain on the overall policy (CMDPs), constrain individual states (state wise constraints), constrain actions, etc…
> >
> > For instance, CMDPs can arise as alternatives to reward shaping: For example, instead of trying to accurately weight the contribution of walking over the red squares and arriving at the goal using some uninterpretable scalars $\alpha_{safety}r_{square} + \alpha_{reachable}r_{goal}$, it is replaced by a much easier to handle explicit statement where one adds a direct inequality.
> > We only discuss the first, while the others – while interesting – do not really connect with the CMDP problem that is solved in this paper.
> >
> > Or maybe you are thinking of model-based methods? Of course any e.g. MPC based approach can integrate changes in the constraints since they directly produce a model of the dynamics, but the objective of our work is to specifically look at CMDPs without need for a model of the environment.
> >
> > Do you think it makes sense to expand our related work section to other kinds of constraints?

---

> > > ### Comment · Reviewer_peX1 · 2026-04-01
> > >
> > > Thanks the authors for the reply, the current revision has resolved most of my concerns. The only limitation of this paper is the restricted evaluation of the proposed method, which is also mentioned by the other reviewers in the discussion.

---

### Review · Reviewer_nhiT · 2026-03-02

**Summary Of Contributions:**

The paper deals with safe online RL, where safety is considered in the sense of a CMDP. A new approach is presented that utilizes the successor representation. A particular advantage of the method is stated to be that the policy can be quickly adapted to changes in the reward and/or cost function without complete re-training.

**Additional Comments:**

I find it somewhat unsatisfactory that Figure 4 does not indicate the uncertainty of the results. However, this is not entirely trivial, as with the statistics of counts presented here, a simple standard deviation of the mean, while better than nothing, probably does not provide realistic uncertainties for counts near 0. One could probably arrive at good estimates of the uncertainties using Bayesian inference of a binomial (with the beta distribution as a conjugate prior and a flat prior with alpha=1 and beta=1).

To what extent can the method be extended to continuous state spaces?

The limitations of the current approach should be presented more clearly.

**Audience:**

Yes

**Audience Explanation:**

I think the topic is very interesting. Safety and fast adaptability are qualities that are important when using RL in real-world systems.

**Broader Impact Concerns:**

no concerns

**Claims And Evidence:**

Yes

**Claims Explanation:**

It is shown, using a specially created benchmark in various configurations, that the method fundamentally works, that it performs better in terms of safety than some baseline methods, and is even best in the 'three-rooms' configuration. It is also shown that the method reacts to changed rewards and costs.

**Requested Changes:**

A particular advantage of the method is stated to be that the policy can be quickly adapted to changes in the reward and/or cost function without complete re-training. Therefore, it is absolutely essential to mention in the text that both the desire for fast adaptability and the solution approach by decoupling dynamics and reward/cost or reward components have already been published in [1].
This does not diminish the novelty of the approach presented here, as there are very clear technological differences; for example, the method in [1] is model-based and without an explicit representation of the Q-function.

Furthermore, for a journal publication, I consider it necessary to perform at least ten repetitions and not just five, as done here.

In the bibliography, there are probably unintended lowercase spellings, such as 'lagrangian'.

[1] Marc Weber et al., „Learning control policies for variable objectives from offline data“, 2023

---

> ### Author Response · Authors · 2026-03-20
>
> Thank you for your review!
> We are already running additional seeds for better statistics. We will also add the Model based paper you mentioned: In general, we focused our prior work section on model-free methods since the adaptation to new cost functions is not difficult in e.g. a Model Predictive Control setting. Here, one just has to deal with the immediate costs and can deal with the temporal aspect via online planning.
>
> Regarding uncertainty: We can add the standard deviation to the plot. This should be sufficient since for large N (=number of rollouts) the Bernoulli distribution approaches a normal one.
>
> >To what extent can the method be extended to continuous state spaces?
>
> We already assume a continuous state space. Specifically, our actions consist of picking the cardinal direction (done by the policy) which is then executed with a normally distributed step length. This is to add stochasticity to the model by considering action noise caused by e.g. inaccurate update frequency or suboptimal motors.
>
> The true limitation is towards continuous action spaces: These are technically still modellable by considering the online optimization of argmax_a Q(a,s), but in practice the optimization over a is too hard to do online. This is what gives rise to methods like DDPG or SAC, which learn a policy to amortize finding the “argmax action”. Our method should plug into all of these methods directly, since they just assume the existence of Q(a,s), not any specific parameterization.
>
> We will highlight the limitations of this approach more clearly: Specifically, the core assumption of all “Successor representation” models is that the features are “sufficiently nice” to allow for a linearization of the reward. This is not generally hard to do in real world situations (since you know the task you want to solve), but is not given by general “unknown” CMDPs (see also our answer to 5FbN)
> Do you think making this limitation explicit is sufficient for this?

---

### Review · Reviewer_5FbN · 2026-03-16

**Summary Of Contributions:**

This paper proposes SafeDSR, which applies deep successor representations to constrained MDPs. The key idea is that successor features decouple environment dynamics from reward and cost estimation, so that when the cost function changes, only a linear cost weight matrix needs retraining rather than the full network. The paper introduces a SafeDQN baseline, proposes the "safe goal count/rate" as evaluation metrics, and evaluates on a continuous 2D grid world with configurable cost regions. The authors demonstrate competitiveness with existing CRL methods and show adaptation to novel cost distributions.

### Strengths:

- The main idea is intuitive. Using successor representations to separate dynamics from reward and cost is a natural fit for settings where the cost map changes but the transition dynamics do not.
- The adaptation experiments are a useful proof of concept. The paper does show that the learned representation can be reused when the cost distribution changes, which is the main motivation of the work

### Weaknesses:

- The main metric, safe goal count / safe goal rate, is not the quantity optimized by the CMDP formulation given in the paper. The formulation is expectation-based over discounted reward and discounted cumulative cost, while safe goal count is a binary zero-violation metric.
- Evaluation is limited to a single 2D grid world with low-dimensional state and discrete actions. While useful for interpretability, this is insufficient to establish the method's viability. Standard CRL benchmarks (Safety-Gymnasium, SafetyRL suites) use high-dimensional observations and continuous actions. The scalability of the successor feature decomposition to settings where the feature encoder $\phi$ must handle complex inputs is unexamined.

**Audience:**

Yes

**Audience Explanation:**

The intersection of successor representations and constrained RL is underexplored and the adaptation angle is practically relevant. However, the limited evaluation and incremental novelty reduce the impact.

**Claims And Evidence:**

No

**Claims Explanation:**

The claims are explored partially but not with sufficient evidence.

- The "safe goal count" metric is not what the other algorithms are optimized for and counts trajectories with zero constraint violations. This is a much stricter criterion closer to almost-sure safety or chance constraints, not the expectation-based CMDP formulation the paper claims to solve. An agent that occasionally incurs a low cost per episode (well under budget) but reliably reaches the goal would score poorly on "safe goal count" despite being an excellent CMDP solution.
- While ignoring the hyperparameter search over baselines is mentioned, simply testing 11 algorithms and claiming that this already covers a wide parameter range is ill founded, because each algorithm has its own sensitivity profile. A fairer protocol would be to do at least a coarse grid search for each baseline, or to use the baselines' own recommended parameters for discrete-action gridworld settings (rather than defaults that may be tuned for continuous MuJoCo tasks).

**Requested Changes:**

1. (Critical for acceptance) In the absence of a deep theoretical analysis the paper needs proof of scalability beyond simplistic environments. At minimum, the paper should include a more realistic safe RL benchmark such as Safety-Gymnasium, or clearly narrow its claims to toy navigation tasks. The paper currently claims strong performance while using a benchmark it explicitly describes as not representative of modern RL challenges.
2. (Critical for acceptance) Report expected episodic return and expected episodic cost alongside safe goal count/rate. If the authors wish to advocate for hard-constraint satisfaction as the primary objective, the problem should be reframed away from the CMDP formulation and discussed in relation to shielding and constraint-violation-reducing approaches such as weakest-precondition-guided exploration [1] and worst-case constrained methods [2, 3].
3. (Would strengthen the work) Using a fairer baseline protocol. Either tune the baselines with a comparable budget, or significantly soften the claims about outperforming existing methods. The current setup gives SafeDSR an advantage that is acknowledged in the paper.

### References:

[1] Guiding Safe Exploration with Weakest Preconditions, Anderson et al., 2023

[2] WCSAC: Worst-Case Soft Actor Critic for Safety-Constrained Reinforcement Learning, Yang et al., AAAI 2021

[3] Robust Adaptive Multi-Step Predictive Shielding, Ambadkar et al., ICLR 2026

---

> ### Author Response · Authors · 2026-03-20
>
> First and foremost we want to thank the reviewer for reading our work and noting the strength of the adaptation freedom our framework gives.
> Regarding your concerns:
> > The "safe goal count" metric is not what the other algorithms are optimized for and counts trajectories with zero constraint violations
>
> We were perhaps a little unclear in our definition of the “safe goal count”:
> Reaching the goal safely means that we do not violate the constraints, i.e. Cost(trajectory) <= budget. It does not mean we reach the goal with zero costs. This means that of the agent e.g. cuts a single corner, the trajectory might still be considered safe, iff the cost remains below the budget.
> Essentially, the safe goal count is estimating p(goal reached, cost<budget). This is generally a less restrictive metric since it clamps the long tails of trajectories with cost>>budget.
> However, adding the actual cost/reward should not be difficult. The reason we did not do so initially is mostly due to the complexity of comparing different (perhaps unsafe) agents on changing environments, which generally affects the e.g. total reachable reward/cost. The safe goal metric is consistent even when changing the environment, which is exactly the subject of this paper.
>
> > While ignoring the hyperparameter search over baselines is mentioned, simply testing 11 algorithms and claiming that this already covers a wide parameter range is ill founded, because each algorithm has its own sensitivity profile
>
> This is true, however, the policies were tuned on a substantially similar environment (safety gyms “Safety[Agent]Goal” task), which is equally a goal reaching task with similar dynamics. Please also note that our environment is not a discrete gridworld: The state is continuous. Specifically, the actions choose one of the cardinal directions, after which a step using a random stepsize is performed (see section 5.1). This corresponds to performing the navigation problem under uncertainty (e.g. inexact motion control or suboptimally discretized time).
> In general, one of the fundamental limitations of any Successor Representation model is the need for good state representations, since otherwise a linear decomposition in the form of (7) does not exist. If such a representation does not exist, the problem becomes essentially partially observable since states can no longer be distinguished by the reward/successor function. This was also noted by [1], who highlight the necessity of good features for successor representation:
> > One of the major issues with the DSR is learning discriminative features. In order to scale up our approach to more expressive environments, it will be crucial to combine various deep generative and self-supervised models […]
>
> I.e. this is a limitation that goes beyond our work and hits the core of Successor Representation research.
> Our work is not a fundamental improvement of SRs, but rather an extension making use of existing SR methods.
> Learning this representation from scratch is notoriously difficult since one runs into the issue that spaces that are not sampled are not captured in the features, which leads to them not being selected (hencewhy prior work highlights the need for unsupervised learning methods).
>
> In real world environments where one generally knows about the problem, this is only a small assumption - For instance [2] (appendix B.1.2) just places Radial basis functions on evenly spaced gridpoints to ensure their 2d environment is linearizable - but this is not generally the case for e.g. safety gymnasium. One could design such a state representation, but then we wouldn’t really be benchmarking on the safety gymnasium environments anymore. Successor representations generally value different things in features compared to “normal” RL (normal RL wants compact representations, Successor Representations want maximally descriptive representations but care less about compactness). This is why prior work on Successor Representations ([1,2]) also benchmarks in grid-like worlds rather than using standard RL benchmarks.
>
> Our benchmark is inspired by the “SafetyPointGoal” benchmark from a “top down” perspective, where the state is given not as relative distances to the danger areas, but rather as absolute coordinates (and, of course, with the ability to pick and choose fixed constrained areas allowing us to evaluate environmental changes).
> We will clarify that our model does assume a suitable feature representation exists.
>
>
> [1] Tejas D. Kulkarni, Ardavan Saeedi, Simanta Gautam, and Samuel J. Gershman. Deep successor reinforcement learning. ArXiv, abs/1606.02396, 2016
>
> [2] Andre Barreto, Will Dabney, Remi Munos, Jonathan J Hunt, Tom Schaul, Hado P van Hasselt, and David Silver. Successor features for transfer in reinforcement learning. In I. Guyon, U. Von Luxburg, S. Bengio, H. Wallach, R. Fergus, S. Vishwanathan, and R. Garnett (eds.), Advances in Neural Information Processing Systems, volume 30. Curran Associates, Inc., 2017.

---

### Decision · Action_Editor_stFF · 2026-04-26

**Recommendation:** Accept with minor revision

**Additional Comments:**

The manuscript studies reinforcement learning in settings where safety constraints matter and the problem may change. It proposes Safe Deep Successor Representation (SafeDSR), a method that enables fast policy re‑training by decoupling dynamics, reward structure, and costs through a learnable weight matrix. The approach is introduced and empirically evaluated in a freely configurable continuous‑grid environment. The manuscript does not provide comprehensive theoretical analysis of the method.

Three reviews have been collected. All reviewers see positive aspects in this manuscript. Key contributions and **strengths** of this work include:

* (S1) The topic of efficiently adapting to new constraints and changed environments is a topic of significant interest in Safe RL
* (S2) The manuscript presents an intuitive main idea
* (S3) The empirical adaptation experiments are a useful proof of concept


In their response to the reviewers and their revision of the manuscript, the authors have addressed several concerns by the reviewers.  After the rebuttal, however, still some **weaknesses** pointed out by the reviewers remain:

* (W1) The evaluation is considered still limiting, especially with regards to scale and diversity of problems [peX1, 5FbN]. Reviewer peX1 points out that at least some simple extended experiments would be desirable.  5FbN sees not just an exprimental omission, but potential lack of capturing relevant problem aspects in Safe RL.
* (W2) The authors should reconsider the claims about general competitiveness regarding contrained RL and possibly narrow claims down, especially if no further empirical results can be included. For example, 5FbN suggests framing "the work as a constrained successor-representation proof of concept." [5FbN]
* (W3) Some concerns about main metric (safe goal count) remain, as it does not seem to match the paper’s stated CMDP objective. [5FbN]

Furthermore, the abstract can be improved:
* (W4) From the abstract, it does not become clear what the authors mean by "decoupling of dynamics, reward structure and costs". In the typical RL problem formulation, these components are commonly decoupled (transition dynamics p(s'|s,a) and reward function r(s,a)). The authors are presumably referring to the result of RL/the policy, not the problem formulation.  Still, the novelty isn't fully obvious from the abstract, which should be addressed.  Related to this, clarification on what is cost and what is reward would be helpful (or using only one term if they are meant to be synonymous).

**Summary of evaluation:** Based on the reviews and the outcome of the discussion phase, the manuscript considers an interesting problem and might be suitable for publication in TMLR. However, the authors should consider the weaknesses that remain after the revision and try to address them as much as possible. In particular, they should consider narrowing claims appropriately as also suggested and recommended, especially acknowledging the somewhat limited scope of the empirical results. Please provide a revision with clearly marked changes and response to weaknesses and requests.

**Audience:**

Yes

**Audience Explanation:**

All reviewers agree that the topic is of substantial interest.

**Claims And Evidence:**

Yes

**Claims Explanation:**

This is a partial "yes".  Not all reviewers are fully convinced by the claims and evidence.  In particular, the empirical results have been pointed out as limiting (see weaknesses below).  In their revision, the authors decided to do additional experimental runs, but did not include additional problems/examples.  This can be acceptable if the authors adjust their claims (e.g., it has been suggested to frame the empirical results more as a proof-of-concept).  This issue can probably be fixed in a minor revision.